# Harnessing Untrained Dynamics: A Reservoir Computing Approach to State-Space Models

## Abstract

We introduce the Reservoir State Space Model (RSSM), a novel neural architecture that integrates the structured dynamics of State Space Models (SSMs) with the efficiency of reservoir computing to address long-term dependencies in sequence modeling. Leveraging the linear structure of SSMs, RSSMs implement efficient convolutional operations that maintain a latent internal state, akin to Recurrent Neural Networks (RNNs), while enabling fast and parallelizable computation. We conduct a stability analysis of the underlying SSMs to extend the memory capacity of the model, ensuring rich and expressive hidden representations.

A key innovation of RSSM is its use of untrained, structured convolutional dynamics as a fixed reservoir, with learning confined to a lightweight feed-forward readout layer. This design drastically reduces training complexity and computational overhead, making RSSMs well-suited for low-resource or real-time applications. Empirical evaluations on standard sequence modeling benchmarks demonstrate that RSSMs achieve competitive accuracy while offering significant efficiency gains compared to traditional trainable architectures. Our results establish RSSMs as a new class of sequence models that combines the strengths of structured, fixed dynamics with the flexibility of learned representations, offering a compelling trade-off between performance and efficiency.

## 1 Introduction

Efficiently modeling long-term dependencies in sequential data remains a major challenge in machine learning Bengio et al. (1994). This issue is prominent in tasks such as text summarization and sentiment analysis Cho et al. (2014), where understanding depends on information introduced much earlier in the sequence. Similar challenges appear in audio processing, where correct interpretation relies on long-range context.

While transformers have proven effective across many sequence modeling tasks, their quadratic complexity with respect to sequence length makes them computationally expensive for long inputs Vaswani et al. (2017). To address these limitations, structured state-space models (SSMs) have emerged as a promising alternative Gu et al. (2021); Nguyen et al. (2022), leveraging their equivalence with convolutions to enable parallel computation and faster training while maintaining an internal state akin to RNNs.

In this work, we explore the integration of reservoir computing Lukoševičius & Jaeger (2009); Nakajima & Fischer (2021) with SSMs to define a novel, convolution-based reservoir model. Our approach utilizes untrained convolutional layers to form a deep reservoir that captures rich temporal features, inspired by principles of deep reservoir computing Gallicchio et al. (2017). Only a feedforward readout layer is trained. This improves efficiency and reduces the need for extensive parameter updates. This architecture offers long short-term memory, computational efficiency, and sustainability benefits. It lowers training costs and energy consumption, supporting more environmentally friendly AI practices Strubell et al. (2019) and increasing accessibility for institutions with limited resources.

The contributions of this paper are:

- **Integration of SSMs into the Reservoir Computing framework**. We propose Reservoir State-Space Model (RSSM), an efficient reservoir computing model that structures the recurrent dynamics following principles of modern SSMs.

- **Stability analysis of RSSM.** We establish sufficient conditions for the stability of RSSM from both a classical recurrent perspective and a convolutional interpretation, offering complementary insights into their dynamic behavior.

- **Excellent Efficiency-Accuracy trade-off.** We benchmark RSSM against a diverse set of sequence modeling approaches, including reservoir computing models, fully-trainable RNNs, efficient Transformer variants, and the state-of-the-art S4 architecture. RSSM demonstrates a compelling balance between computational efficiency and predictive accuracy, highlighting its competitiveness across a range of sequence modeling tasks.

## 2 RELATED WORKS

**Recurrent Neural Networks.** Recurrent Neural Networks (RNNs) are foundational for sequence modeling due to their temporal recurrence and memory Mikolov et al. (2010), but struggle with vanishing/exploding gradients on long dependencies Bengio et al. (1994); Pascanu et al. (2013). Gated variants like LSTMs Hochreiter & Schmidhuber (1997) and GRUs Cho et al. (2014); Chung et al. (2014) address this via gating mechanisms. Further improvements leverage norm-preserving parameterizations such as unitary matrices Arjovsky et al. (2016), Householder/Givens rotations Mhammedi et al. (2017); Jing et al. (2017), and Cayley transforms Helfrich et al. (2018). Recent models like xLSTM Beck et al. (2024) scale memory via exponential gating and matrix structures, enhancing long-term retention in deep architectures.

**Reservoir Computing.** Reservoir Computing (RC) provides a training-efficient alternative to RNNs by leveraging fixed high-dimensional dynamical systems and training only a linear readout Lukoševičius & Jaeger (2009); Nakajima & Fischer (2021). Prominent models include Echo State Networks (ESNs) Jaeger (2007) and Liquid State Machines Maass (2011). RC's simplicity and computational efficiency make it attractive for resource-constrained applications. Deep variants, such as DeepESNs Gallicchio et al. (2017), have been developed to model hierarchical temporal features, enabling richer multi-timescale dynamics without full backpropagation Gallicchio et al. (2018).

**Transformers and Efficient Attention.** Transformers Vaswani et al. (2017) dominate sequence modeling through self-attention, which enables global context aggregation and parallelism. However, their quadratic complexity in sequence length limits scalability. Efficient variants—such as Sparse Transformers Child et al. (2019), Longformer Beltagy et al. (2020), BigBird Zaheer et al. (2020), Linformer Wang et al. (2020), and Reformer Kitaev et al. (2020)—introduce sparsity, low-rank approximations, and locality to reduce this cost. Some alternatives, like Synthesizer Tay et al. (2021) and linear attention models Katharopoulos et al. (2020), question the necessity of self-attention entirely. Despite these advances, modeling long-range dependencies remains a fundamental challenge Tay et al. (2020); Wen et al. (2022).

**Deep State Space Models.** State Space Models (SSMs) offer a compelling alternative by learning continuous-time latent dynamics for sequence modeling. The HiPPO framework Gu et al. (2020) introduces a memory-efficient representation, inspiring deep variants like S4 Gu et al. (2022a), which uses diagonal plus low-rank parameterizations and FFT-based computation to replace attention. Simplified models such as DSS and S4D Gupta et al. (2022); Gu et al. (2022b) retain efficiency while preserving performance. These models generalize well across domains including language Mehta et al. (2022), vision Nguyen et al. (2022), and audio Goel et al. (2022), showing promise for long-sequence processing.

## 3 METHOD

In this section, we introduce the proposed method, detailing its architecture and underlying dynamics. We analyze its stability, offer a convolutional interpretation, and discuss the benefits of depth, concluding with a comparison of its computational complexity to state-of-the-art models.

### 3.1 RSSM BLOCK

The foundational block of our structured reservoir computing approach is based on linear state space models. We briefly outline the foundational concepts required for our methodology, and refer to Appendix D for more details. A linear continuous SSM is a parameterized linear time-invariant system defined by

$$\begin{cases} \mathbf{x}'(t) = \mathbf{A}\mathbf{x}(t) + \mathbf{B}u(t) \\ y(t) = \Re\left(\mathbf{C}\mathbf{x}(t)\right) + \mathbf{D}u(t), \quad t \in \mathbb{R}_+. \end{cases} \tag{1}$$

where $\mathbf{x}(t) \in \mathbb{C}^P$ is the internal state, $u(t) \in \mathbb{R}$ is the input, $y(t) \in \mathbb{R}$ is the output. The model parameters are the state matrix $\mathbf{A} \in \mathbb{C}^{P \times P}$, the input matrix $\mathbf{B} \in \mathbb{C}^{P \times 1}$, the output matrix $\mathbf{C} \in \mathbb{C}^{1 \times P}$, the skip connection $\mathbf{D} \in \mathbb{R}$, and $\Re(\mathbf{z})$ denotes the real part of a complex-valued vector $\mathbf{z}$. Through discretization, a discrete SSM is obtained as follows.

$$\begin{cases} \mathbf{x}_k = \overline{\mathbf{A}}\mathbf{x}_{k-1} + \overline{\mathbf{B}}u_k \\ y_k = \Re\left(\overline{\mathbf{C}}\mathbf{x}_k\right) + \overline{\mathbf{D}}u_k, \quad k = 0, \dots, L-1 \end{cases} \tag{2}$$

where the new parameters $\overline{\mathbf{A}}, \overline{\mathbf{B}}, \overline{\mathbf{C}}, \overline{\mathbf{D}}$ are obtained by a discretization method, e.g., ZOH (Definition D.3), using a constant sampling frequency $\Delta$. The linear discrete SSM admits an efficient representation as a convolution, which we exploit to build the RSSM block.

Our proposed RSSM block is a sequence-to-sequence model that efficiently maps an input time series $\mathbf{u}$ to an output time series $\mathbf{y}$ of the same length. Specifically, given a linear discrete SSM, the kernel $\overline{\mathbf{K}} = \left[\overline{\mathbf{C}}\overline{\mathbf{B}}, \dots, \overline{\mathbf{C}}\,\overline{\mathbf{A}}^l\overline{\mathbf{B}}, \dots, \overline{\mathbf{C}}\,\overline{\mathbf{A}}^{L-1}\overline{\mathbf{B}}\right] \in \mathbb{C}^{1 \times L}$, where $L$ is the sequence length, defines the linear convolution mapping from the input sequence $\{u_k\}_{k=0}^{L-1} = \mathbf{u} \in \mathbb{R}^{1 \times L}$ to the output sequence $\{y_k\}_{k=0}^{L-1} = \mathbf{y} \in \mathbb{R}^{1 \times L}$, which takes the following form $y_k = \left(\Re\left(\overline{\mathbf{K}} * \mathbf{u}\right) + \overline{\mathbf{D}}\,\mathbf{u}\right)_k = \Re\left(\sum_{l=0}^{k} \overline{\mathbf{K}}_l\, u_{k-l}\right) + \overline{\mathbf{D}}\,u_k$. The linear convolution is efficiently computed using the discrete Fourier transform (DFT) and the Convolution Theorem Oppenheim & Schafer (2010):

$$\mathbf{y} = \Re\left(DFT^{-1}\left(DFT\left(\overline{\mathbf{K}}\right) \odot DFT\left(\mathbf{u}\right)\right)\right) + \overline{\mathbf{D}}\,\mathbf{u}. \tag{3}$$

To enhance computational efficiency, we adopt a diagonal discrete state transition matrix. Beyond reducing parameter count and model complexity, this design choice also enables a more tractable theoretical analysis of RSSM stability, as detailed in Section 4. Given a linear continuous SSM with diagonal complex state matrix $\mathbf{A} = \mathrm{diag}(\lambda_0, \dots, \lambda_{P-1})$, the diagonal discrete state matrix after the Zero-Order Hold (ZOH) transform with sampling interval $\Delta > 0$ is:

$$\overline{\mathbf{A}} = e^{\Delta \mathbf{A}} = \mathrm{diag}(\rho_0 e^{i\theta_0}, \dots, \rho_{P-1} e^{i\theta_{P-1}}) \tag{4}$$

where $\rho_i = e^{\Delta \Re(\lambda_i)}$ and $\theta_i = \Delta \Im(\lambda_i)$ for $i = 0, \dots, P-1$. To enable an RSSM block to process multivariate inputs, we incorporate $H$ linear SSMs in parallel within a single block. Now, since we assume diagonal state transition matrices of dimension $P$, we can parameterize an RSSM block by the state matrix $\mathbf{A} \in \mathbb{C}^{P \times H}$, sampling rates $\mathbf{\Delta} \in \mathbb{R}^H$, input matrix $\mathbf{B} \in \mathbb{C}^{P \times H}$, output matrix $\mathbf{C} \in \mathbb{C}^{H \times P}$, and skip connection $\mathbf{D} \in \mathbb{R}^H$, collectively defining $H$ independent continuous SSMs. The state matrix $\mathbf{A}$ is initialized with eigenvalues $\lambda_{i,j}$ such that $\Re(\lambda_{i,j}) \in [m_\mathbf{A}, M_\mathbf{A}]$ and $\Im(\lambda_{i,j}) \in [0, 2\pi)$. The sampling rates control the discretization process and are $\mathbf{\Delta} \in [m_\Delta, M_\Delta]^H$, where $M_\Delta \geq m_\Delta > 0$. The input and output matrices $\mathbf{B}$ and $\mathbf{C}$ are constrained such that their magnitudes lie in $[m_\mathbf{B}, M_\mathbf{B}] \subset \mathbb{R}^+$ and $[m_\mathbf{C}, M_\mathbf{C}] \subset \mathbb{R}^+$ respectively. The skip connection is set such that $\mathbf{D} \in [m_\mathbf{D}, M_\mathbf{D}]^H \subset \mathbb{R}$. This parameterization enables precise regulation of the system's dynamic behavior across all $H$ features while maintaining stability and expressive representations (Sections 4.1 and 4.2). Appendix C provides further details of the RSSM block, including its parameterization and initialization scheme. Our RSSM block does not include a mixing layer that combines the $H$ features at each step of the output sequence generated by the convolutional layer. Learning how the features depend on each other is left to the readout component of the overall architecture, which operates in a high-dimensional space. This choice is motivated by our ablation study on untrained mixing layers in S4D, reported in Appendix A.

### 3.2 RESERVOIR STATE SPACE MODEL (RSSM)

Our approach is inspired by RC and SSMs; accordingly, we aim to intelligently structure the internal connections and leave them untrained after initialization. We refer to this approach as the *RSSM*.

The architecture comprises multiple stacked RSSM blocks, enabling the construction of hierarchical representations. The structure is depicted in Figure 1.

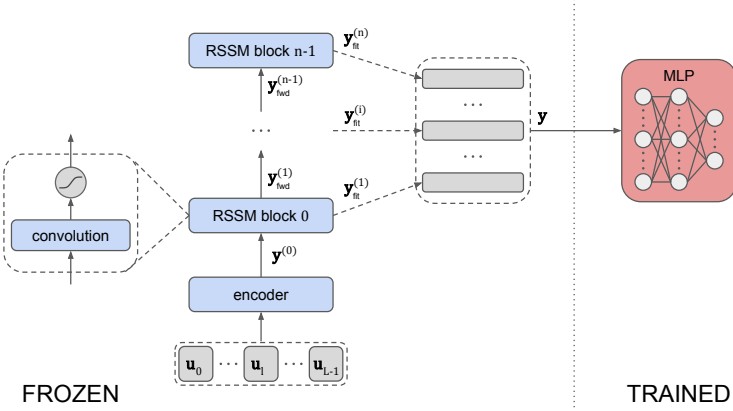

Figure 1: RSSM graphical representation. The batched input sequence is first projected into a higher-dimensional space using an encoder. It then passes through multiple convolutional layers within the RSSM (Equation 5). The activated output from each layer is collected and processed to fit a readout model (Equation 6), enabling efficient sequence modeling and prediction.

The input sequence $\mathbf{u} \in \mathbb{R}^{H_0 \times L}$ is first transformed into a higher-dimensional space, $\mathbb{R}^H$, using an encoder matrix $\mathbf{W}_{en} \in \mathbb{R}^{H \times H_0}$. This higher-dimensional representation is then processed sequentially by a stack of $n$ convolutional layers. We use two distinct activation functions: $\phi_{fwd}(\cdot)$ and $\phi_{fit}(\cdot)$. The activation function $\phi_{fwd}(\cdot)$ is used during the forward pass through the layers of the reservoir model, while $\phi_{fit}(\cdot)$ prepares the layer outputs for the fitting of the readout model. Specifically, we apply the activation function $\phi_{fwd}(\cdot)$ to the output of each layer, and this activated output serves as the input to the next layer (Equation 5).

$$\begin{cases} \mathbf{y}^{(0)} = \mathbf{W}_{en} \cdot \mathbf{u} \\ \mathbf{y}^{(i)} = \Re\left( DFT^{-1}\left( DFT\left(\overline{\mathbf{K}}^{(i)}\right) \odot DFT\left(\mathbf{y}_{fwd}^{(i-1)}\right)\right)\right) + \overline{\mathbf{D}}^{(i)}\mathbf{y}_{fwd}^{(i-1)} \\ \mathbf{y}_{fwd}^{(i)} = \phi_{fwd}\left(\mathbf{y}^{(i)}\right) \\ \mathbf{y}_{fit}^{(i)} = \phi_{fit}\left(\mathbf{y}^{(i)}\right), \quad \text{if } i > 1, \end{cases} \tag{5}$$

where $\overline{\mathbf{K}}^{(i)}$ is the kernel of the $i$-th block, and $\overline{\mathbf{D}}^{(i)}$ is the skip connection of the $i$-th block. Therefore, the output of each layer $\mathbf{y}^{(i)}$ is computed using the convolution view of a discrete SSM. Meanwhile, the outputs of each layer activated by $\phi_{fit}(\cdot)$ are stacked row-wise to produce the global output $\mathbf{y}$ of our reservoir model, as illustrated in Figure 1, and described by the following equation:

$$\mathbf{y} = \begin{bmatrix} \mathbf{y}_{fit}^{(1)} \\ \vdots \\ \mathbf{y}_{fit}^{(n)} \end{bmatrix} \in \mathbb{R}^{nH \times L}. \tag{6}$$

The global output $\mathbf{y}$ produces a hierarchical representation of the input signal in a high-dimensional space. RSSM produces a multiple frequency representation via the global output $\mathbf{y}$ (Equation 6) where progressively higher layers focus on progressively higher frequencies (similar to deep reservoirs in Gallicchio et al. (2017)), thereby increasing the input bandwidth (Section 4.3). We prove in Section 4.3 that this representation is more informative than the original input signal.

The activation function $\phi_{fit}(\cdot)$ does not influence the forward pass of the input sequence through the model's layers, as its role is to fit the readout with activated output. Instead, the choice of $\phi_{fwd}$ is crucial for effectively propagating information through the layers (Section 4.3). We use $ReLU(\cdot)$ for $\phi_{fwd}(\cdot)$ to ensure effective propagation of information through the layers (Theorem 4.2), and $\tanh(\cdot)$ for $\phi_{fit}(\cdot)$ to prepare the output for the readout fitting.

**Efficiency.** For computational efficiency, the RSSM utilizes GPU parallelization for convolutions, moving outputs temporarily to RAM before batch-wise training of a readout model. Given a dataset of $N$ sequences, the readout is an MLP or ridge regression, solving the optimization problem $\mathbf{W}_{out} = \arg\min_{\mathbf{W}} \left\{ \|\mathbf{Y} \cdot \mathbf{W} - \mathbf{d}\|^2 + \alpha \|\mathbf{W}\|^2 \right\}$, where $\mathbf{Y} \in \mathbb{R}^{N \times nH \times L'}$ is the aggregated RSSM output, $\mathbf{d} \in \mathbb{R}^{NL'}$ is the vector of labels, and $\alpha$ the $l$-2 regularization factor. For further details on the training pipeline, see Appendix B.

## 4 THEORETICAL ANALYSIS

We now present a series of theoretical results that establish the stability and representational capacity of our approach. All the proofs are provided in Appendix E.

### 4.1 DYNAMICAL STABILITY

The stability of an RSSM block is determined by the stability of the continuous and discrete SSMs it contains, which in turn depends on their respective state matrices, $\mathbf{A}$ and $\overline{\mathbf{A}}$. These matrices govern the autonomous state dynamics in the absence of external inputs (i.e., when $u(t) = 0$ or $u_l = 0$).

We summarize in Theorem 4.1 the conditions for stability of a deep RSSM architecture.

**Theorem 4.1** (Stability Conditions). *A deep RSSM architecture is stable if all of its constituent RSSM blocks are stable. An RSSM block is stable if and only if each of the $H$ discretised SSMs is stable. A continuous SSM is stable if and only if $\Re(\lambda_i) \leq 0$, and equivalently, a discretized SSM is stable if and only if $\rho_i \leq 1$.*

From Theorem 4.1, the sets of stable state matrices are:
$$\mathcal{S}_{\mathbf{A}} = \left\{ \mathbf{A} \in \mathbb{C}^P \mid \Re(\lambda_i) \leq 0 \right\}, \quad \mathcal{S}_{\overline{\mathbf{A}}} = \left\{ \overline{\mathbf{A}} \in \mathbb{C}^P \mid |\lambda_i| \leq 1 \right\}.$$
The boundaries represent the stability margins:
$$\partial \mathcal{S}_{\mathbf{A}} = \left\{ \mathbf{A} \mid \Re(\lambda_i) = 0 \right\}, \quad \partial \mathcal{S}_{\overline{\mathbf{A}}} = \left\{ \overline{\mathbf{A}} \mid |\lambda_i| = 1 \right\}.$$

To achieve stable yet expressive reservoir dynamics, state matrices are chosen close to these boundaries Ceni & Gallicchio (2024a;b). Specifically, we initialize eigenvalues $\lambda_i$ within:
$$\{\mathbf{A} \mid m_{\mathbf{A}} \leq \Re(\lambda_i) \leq M_{\mathbf{A}} \leq 0, \quad 0 \leq \Im(\lambda_i) < 2\pi\}.$$
Upon discretization, eigenvalues of $\overline{\mathbf{A}}$ have radii $\rho_i \in [e^{\Delta m_{\mathbf{A}}}, e^{\Delta M_{\mathbf{A}}}]$ and angles $\theta_i \in [0, \Delta \cdot 2\pi)$.

The sampling interval $\Delta$ critically impacts internal dynamics by controlling the eigenvalue radii and angles. As $\Delta \to 0$, eigenvalues approach the stability margin ($\rho_i \to 1$, $\theta_i \to 0$), enhancing memory capacity and discretization accuracy. However, excessively small $\Delta$ leads to nearly static dynamics, impairing system responsiveness. Thus, appropriate selection of the hyperparameters ($m_{\mathbf{A}}, M_{\mathbf{A}}, \Delta$) is essential.

Next, we interpret the RSSM from a convolutional viewpoint by examining the derived kernel $\overline{\mathbf{K}}$.

### 4.2 CONVOLUTION FILTER

We examine how the parameters $\mathbf{A}$, $\mathbf{B}$, $\mathbf{C}$, and $\Delta$ influence the convolution kernel $\overline{\mathbf{K}}$ that determines the pre-activated output of the system (Equation 27). Specifically, we focus on the real-valued filter $\overline{\mathbf{F}} = \Re(\overline{\mathbf{K}})$, which governs the actual computation of the pre-activated output signal of the RSSM block for each feature $i = 0, \ldots, H-1$ of the input signal (see Definition D.6).

**Definition 4.1** (RSSM Convolution). Assume a single-channel case ($H = 1$) with no skip connection ($\overline{\mathbf{D}} = 0$), and let $\mathbf{u} \in \mathbb{R}^{1 \times L}$ be a real-valued input signal. The output of the linear RSSM block is given by:
$$\mathbf{y} = \Re(\overline{\mathbf{K}} * \mathbf{u}) = \overline{\mathbf{F}} * \mathbf{u}, \tag{7}$$
where the real-valued filter $\overline{\mathbf{F}}$ has elements:
$$\overline{\mathbf{F}}_l = \sum_{i=0}^{P-1} \nu_i \rho_i^l \cos(\psi_i + l\theta_i) \tag{8}$$
with $\nu_i$ the amplitude, $\rho_i$ the decay/growth rate, $\theta_i$ the frequency, and $\psi_i$ the phase.

The hyperparameters $(P, \{\lambda_i\}, \Delta, \|\mathbf{B}\|, \|\mathbf{C}\|)$ provide a mechanism for filter design, as the following lemmas demonstrate.

**Lemma 4.1** (State Dimension Effect). *The state dimension $P$ determines the number of oscillatory terms in $\overline{\mathbf{F}}$. Higher $P$ allows for more complex filters by superposition of diverse frequencies.*

Similarly to Theorem 4.1, we can find stability constraints from a convolutional perspective, as stated in Lemma 4.2.

**Lemma 4.2** (Stability Constraint). *The $\Re(\lambda_i)$, affecting the decay/growth rates $\rho_i$, govern both the memory capacity and the stability of the convolution by bounding the filter coefficients:*

$$\left(\sum_{i=0}^{P-1} \nu_i\right) \min_i \{\rho_i\}^l \leq |\overline{\mathbf{F}}_l| \leq \left(\sum_{i=0}^{P-1} \nu_i\right) \max_i \{\rho_i\}^l \tag{9}$$

*where $\min_i\{\rho_i\} = e^{\Delta \min_i\{\Re(\lambda_i)\}}$, and $\max_i\{\rho_i\} = e^{\Delta \max_i\{\Re(\lambda_i)\}}$.*

On the other hand, the weight matrices $\mathbf{B}, \mathbf{C}$ do impact the amplitudes and phase shifts of the convolution filter oscillations, as stated in Lemma 4.3.

**Lemma 4.3** (Parameter Scaling). *The matrices $\mathbf{B}, \mathbf{C}$ control the oscillation amplitudes*

$$\nu_i = |\overline{\mathbf{B}}_i||\overline{\mathbf{C}}_i| \propto |\mathbf{B}_i||\mathbf{C}_i| \in [m_{\mathbf{B}} m_{\mathbf{C}}, M_{\mathbf{B}} M_{\mathbf{C}}], \tag{10}$$

*and the phase shifts*

$$\psi_i = arg(\overline{\mathbf{B}}_i) + (\overline{\mathbf{C}}_i) \in [0, 2\pi). \tag{11}$$

Finally, the hyperparameter $\Delta$ has a crucial role in determining the frequencies and decays of oscillations in the convolution filter, as stated in Lemma 4.4.

**Lemma 4.4** (Sampling Rate Effect). *The discretization step $\Delta$ affects the frequencies and decay:*

$$\theta_i = \Delta \Im(\lambda_i), \qquad \rho_i = e^{\Delta \Re(\lambda_i)}. \tag{12}$$

*Larger $\Delta$ increases oscillation frequency but reduces memory capacity.*

This parametric flexibility enables expression of diverse dynamical behaviors, from smooth responses to complex oscillatory patterns. Section 4.3 next examines how the deep architecture processes these filters through nonlinear transformations.

## 4.3 Representational Richness of Deep RSSM

The deep RSSM architecture generates hierarchical signal representations through stacked linear RSSM blocks interleaved with nonlinear activations (Equation 5, Figure 1). While linear RNNs can approximate shift-invariant linear operators Li et al. (2022), they cannot generate new frequency components. Our architecture overcomes this limitation through activation-induced spectral leakage.

**Theorem 4.2** (Spectral Leakage). *Orvieto et al. (2023) Let $u : \mathbb{R} \to \mathbb{R}$ be a continuous-time signal. Let $R_i = [c_i - r_i, c_i + r_i]$ be the $i$-th region activated by the $ReLU(\cdot)$ applied to $u$. Then*

$$FT(ReLU(u))(\omega) = FT(u)(\omega) * \left[\sum_i 2r_i e^{-i\omega c_i} \frac{\sin(\omega c_i)}{\omega c_i}\right],$$

*where $FT$ denotes the Fourier transform, and $*$ the convolution operation.*

This convolution (Theorem 4.2) introduces new frequency components that are impossible for linear systems. As network depth increases, the input undergoes successive non-linear transformations that progressively scale and shift frequency components. This process expands the effective input bandwidth, leading to the emergence of distinct spectral characteristics at each layer.

Therefore, a suitable selection of the fixed parameters $(\mathbf{A}, \mathbf{B}, \mathbf{C}, \mathbf{D}, \Delta)$ ensures stable signal propagation while enabling complex frequency manipulation. The global output (Equation 6) combines these transformed representations into a rich hierarchical encoding.

## 4.4 COMPUTATIONAL COMPLEXITY ANALYSIS

The RSSM achieves optimal computational efficiency by integrating three key properties: (1) a training cost that is independent of sequence length via an MLP readout, (2) parallelizable operations through convolutional processing, and (3) constant-time recurrent inference. This hybrid design effectively avoids the quadratic scaling bottleneck of attention mechanisms while maintaining expressive capacity.

Convolutional operations, which are central to both S4 and RSSM, benefit from efficient computation in the frequency domain. Specifically, when two sequences of length $L$, denoted $\mathbf{h} = h_0, \ldots, h_{L-1}$ and $\mathbf{u} = u_0, \ldots, u_{L-1}$, are convolved, the operation can be computed as element-wise multiplication in the frequency domain using the Discrete Fourier Transform (DFT), $\mathbf{h} * \mathbf{u} = DFT^{-1}(DFT(\mathbf{h}) \odot DFT(\mathbf{u}))$. This approach yields a time complexity of $\mathcal{O}(L\log(L))$, as established in standard signal processing references Oppenheim et al. (1999); Proakis & Manolakis (1996).

Table 1: Comparison of model complexities across convolutional, recurrent, attention-based, S4, ESN, and our proposed RSSM. The table summarizes key metrics, including the number of trainable parameters, training complexity, reservoir complexity, space complexity, parallelization capabilities, and inference complexity. $\tilde{L}$ and $\tilde{H}$ denote logarithmic factors in $L$ and $H$, respectively. All the cost complexities should be interpreted in $\mathcal{O}(\cdot)$ (big O notation). Bold denotes model is theoretically best for that metric.

|  | Trainable pars | Training | Reservoir | Space | Parallel | Inference |
|---|---|---|---|---|---|---|
| Convolution | $LH$ | $BL\tilde{L}H$ | $-$ | $\mathbf{BLH}$ | **Yes** | $LH^2$ |
| Recurrence | $\mathbf{H^2}$ | $BLH^2$ | $-$ | $\mathbf{BLH}$ | No | $\mathbf{H^2}$ |
| Attention | $\mathbf{H^2}$ | $BLH(L+H)$ | $-$ | $BL(L+H)$ | **Yes** | $L^2H + H^2L$ |
| S4 | $\mathbf{H^2}$ | $BLH(\tilde{L}+\tilde{H})$ | $-$ | $\mathbf{BLH}$ | **Yes** | $\mathbf{H^2}$ |
| ESN | $\mathbf{H^2}$ | $\mathbf{BH^2}$ | $BLH^2$ | $\mathbf{BLH}$ | No | $\mathbf{H^2}$ |
| RSSM *(ours)* | $\mathbf{H^2}$ | $\mathbf{BH^2}$ | $BL\tilde{L}H$ | $\mathbf{BLH}$ | **Yes** | $\mathbf{H^2}$ |

As shown in Table 1, RSSM matches or outperforms existing architectures across all key complexity dimensions. By decoupling readout cost from sequence length, it improves training efficiency while retaining the inference-time advantages of recurrent models. This makes RSSM particularly well-suited for long-sequence modeling scenarios where computational efficiency and expressivity are both essential.

## 5 EXPERIMENTS

In this section, we present the experiments conducted on various tasks to compare the performance of our method against other baseline models from the literature. We evaluate the experiments based on both effectiveness and efficiency in classification tasks. All experiments have been executed on a Tesla V100-PCIE-16GB GPU. The source code for the method, analysis, and all experiments will be made publicly available on GitHub upon acceptance.

Appendix F describes the architecture of our model for classification tasks and the other baseline models. We benchmark our model on the sMNIST, psMNIST Le et al. (2015), sCIFAR-10 tasks Gu et al. (2022a), and the pool of Long Range Arena tasks Tay et al. (2020). Appendix F.1 details the approach used for hyperparameter tuning and selecting optimal model configurations.

### 5.1 PERFORMANCE AND EFFICIENCY ON PIXEL-LEVEL IMAGE CLASSIFICATION

In Table 2 we report results including training time, $CO_2$ emissions (in grams), energy consumption (in kWh), and accuracy. For the reservoir-based models (ESN, DeepESN, RSSM-r, and RSSM), the time refers to the computation of the reservoir output plus the fitting time of the readout. Bold green indicates the best value for a given metric if it corresponds to an RC-based model, whereas bold red indicates the best value if it corresponds to a fully trainable model. The hyperparameter search space for each model is detailed in Appendix G, and the optimal hyperparameters are listed in Appendix H.

Table 2: Performance results on the sMNIST, psMNIST, and sCIFAR-10 datasets.

| | sMNIST | | | |
|---|---|---|---|---|
| | Time (s) | Emissions | Energy | Acc |
| GRU | $2.21_{\pm 0.04} \times 10^3$ | $1.24_{\pm 0.03} \times 10^{-1}$ | $5.49_{\pm 0.06} \times 10^{-1}$ | $99.26_{\pm 0.16}$ |
| S4 | $1.47_{\pm 0.03} \times 10^3$ | $8.38_{\pm 0.07} \times 10^{-2}$ | $3.71_{\pm 0.07} \times 10^{-1}$ | $\textcolor{red}{\mathbf{99.30_{\pm 0.14}}}$ |
| ESN | $1.03_{\pm 0.02} \times 10^2$ | $5.68_{\pm 0.06} \times 10^{-3}$ | $2.51_{\pm 0.04} \times 10^{-2}$ | $66.65_{\pm 0.32}$ |
| DeepESN | $51.82_{\pm 0.31}$ | $3.02_{\pm 0.05} \times 10^{-3}$ | $1.33_{\pm 0.02} \times 10^{-2}$ | $72.92_{\pm 0.35}$ |
| RSSM-r *(ours)* | $\mathbf{47.49_{\pm 0.39}}$ | $\mathbf{2.70_{\pm 0.04} \times 10^{-3}}$ | $\mathbf{1.20_{\pm 0.02} \times 10^{-2}}$ | $97.07_{\pm 0.22}$ |
| RSSM *(ours)* | $91.40_{\pm 0.42}$ | $4.88_{\pm 0.05} \times 10^{-3}$ | $2.16_{\pm 0.03} \times 10^{-2}$ | $98.61_{\pm 0.19}$ |
| | pMNIST | | | |
| | Time (s) | Emissions | Energy | Acc |
| GRU | $1.97_{\pm 0.03} \times 10^3$ | $1.11_{\pm 0.02} \times 10^{-1}$ | $4.89_{\pm 0.03} \times 10^{-1}$ | $94.96_{\pm 0.41}$ |
| S4 | $1.29_{\pm 0.02} \times 10^3$ | $7.32_{\pm 0.09} \times 10^{-2}$ | $3.24_{\pm 0.05} \times 10^{-1}$ | $98.08_{\pm 0.19}$ |
| ESN | $1.03_{\pm 0.01} \times 10^2$ | $5.81_{\pm 0.06} \times 10^{-3}$ | $2.57_{\pm 0.03} \times 10^{-2}$ | $92.76_{\pm 0.31}$ |
| DeepESN | $56.41_{\pm 0.33}$ | $3.52_{\pm 0.04} \times 10^{-3}$ | $1.55_{\pm 0.03} \times 10^{-2}$ | $90.50_{\pm 0.37}$ |
| RSSM-r *(ours)* | $\mathbf{50.91_{\pm 0.37}}$ | $\mathbf{2.84_{\pm 0.02} \times 10^{-3}}$ | $\mathbf{1.26_{\pm 0.02} \times 10^{-2}}$ | $95.59_{\pm 0.24}$ |
| RSSM *(ours)* | $95.73_{\pm 0.55}$ | $5.16_{\pm 0.06} \times 10^{-3}$ | $2.28_{\pm 0.03} \times 10^{-2}$ | $\mathbf{98.10_{\pm 0.20}}$ |
| | sCIFAR-10 | | | |
| | Time (s) | Emissions | Energy | Acc |
| GRU | $1.73_{\pm 0.02} \times 10^4$ | $1.14_{\pm 0.01}$ | $5.03_{\pm 0.06}$ | $73.57_{\pm 0.42}$ |
| S4 | $1.28_{\pm 0.02} \times 10^4$ | $9.12_{\pm 0.07} \times 10^{-1}$ | $4.03_{\pm 0.03}$ | $\textcolor{red}{\mathbf{84.72_{\pm 0.31}}}$ |
| ESN | $3.62_{\pm 0.04} \times 10^3$ | $2.45_{\pm 0.03} \times 10^{-1}$ | $1.08_{\pm 0.02}$ | $34.09_{\pm 0.50}$ |
| DeepESN | $6.99_{\pm 0.05} \times 10^2$ | $4.15_{\pm 0.06} \times 10^{-2}$ | $1.84_{\pm 0.03} \times 10^{-1}$ | $31.76_{\pm 0.39}$ |
| RSSM-r *(ours)* | $\mathbf{5.70_{\pm 0.06} \times 10^2}$ | $\mathbf{3.73_{\pm 0.05} \times 10^{-2}}$ | $\mathbf{1.65_{\pm 0.03} \times 10^{-1}}$ | $60.02_{\pm 0.37}$ |
| RSSM *(ours)* | $1.21_{\pm 0.02} \times 10^3$ | $8.49_{\pm 0.07} \times 10^{-2}$ | $3.75_{\pm 0.05} \times 10^{-1}$ | $63.31_{\pm 0.32}$ |

The results on sMNIST, pMNIST, and sCIFAR-10 (Tables 2) are evaluated using training time, accuracy, $CO_2$ emissions, and energy consumption, see Appendix F.2. RSSM performs competitively with fully trainable models like GRU and S4, while outperforming reservoir methods such as ESN and DeepESN.

On sMNIST and pMNIST, RSSM matches GRU and S4 in accuracy, and notably surpasses them on pMNIST, showing robustness to permuted sequences. Unlike ESN and DeepESN, which suffer from late-timestep zero-padding, RSSM maintains strong performance, underscoring its superior memory capacity.

Table 3: Model speeds (iterations/second) for both training and inference. Higher is better.

| | sMNIST ($L = 784$) | sCIFAR-10 ($L = 1024$) |
|---|---|---|
| **Training** | | |
| RSSM *(ours)* | 342.18 | 14.08 |
| GRU | 7.06 (48.4×) | 1.16 (12.1×) |
| S4 | 12.24 (27.9×) | 2.28 (6.1×) |
| **Inference** | | |
| RSSM *(ours)* | 5.68 | 3.86 |
| ESN | 2.39 (2.3×) | 0.38 (10×) |
| DeepESN | 4.94 (1.4×) | 3.12 (1.2×) |

In efficiency, RSSM excels in both MLP readout training and reservoir computation. Table 3 compares training runtimes, focusing on RSSM's MLP readout speed across tasks (excluding pMNIST, which shares sMNIST's sequence length). RSSM trains significantly faster than GRU and S4 by learning dependencies only at the final timestep of the global output (Equation 6), leveraging reservoir memory (Section 4.4, Table 1).

The Inference runtimes refer to the forward pass (excluding the readout fitting), showing RSSM's reservoir computation is faster than ESN and DeepESN due to its unique parallelization. pMNIST speeds are excluded for sequence length parity.

This efficiency results in lower energy use and $CO_2$ emissions (Tables 2) compared to GRU and S4. On the more complex sCIFAR-10, RSSM maintains superior efficiency with competitive accuracy—outperforming ESN and DeepESN and approaching S4—making it a strong candidate for resource-constrained or energy-aware applications.

## 5.2 Long Range Arena benchmark

We evaluate RSSM and RSSM-r on the LRA benchmark, which tests a model's ability to capture very long-range dependencies across diverse tasks (Table 4). While transformer models perform well on simpler tasks like text classification, they consistently fail on extreme long-range tasks like Path-X, where even efficient variants such as Linformer and Performer underperform. In contrast, RSSM achieves the highest accuracy on Pathfinder and is the only model to succeed on Path-X, demonstrating superior long-sequence modeling through untrained convolutions without incurring the quadratic cost of attention.

Results underscore RSSM's architectural strengths and efficiency in long-term dependency retention, positioning it as a robust alternative to transformers.

Table 4: Accuracy on the LRA benchmark. Transformer and S4 results are from Tay et al. (2020) and Gu et al. (2022a), respectively. Best and second-best scores are in bold and underlined. FAIL indicates transformers failed to learn Path-X.

| Model | Image | ListOps | Text | Pathfinder | Path-X |
|---|---|---|---|---|---|
| Chance | 10.00 | 10.00 | 50.00 | 50.00 | 50.00 |
| Transformer | 42.44 | 36.37 | 64.27 | 71.40 | FAIL |
| Local attention | 41.46 | 15.82 | 52.98 | 66.63 | FAIL |
| Sparse Trans. | 44.24 | 17.07 | 63.58 | 71.71 | FAIL |
| Longformer | 42.22 | 35.63 | 62.85 | 69.71 | FAIL |
| Linformer | 38.56 | 35.70 | 53.94 | 76.34 | FAIL |
| Reformer | 38.07 | 37.27 | 56.10 | 68.50 | FAIL |
| Sinkhron trans. | 41.23 | 33.67 | 61.20 | 67.45 | FAIL |
| Synthesizer | 41.61 | 36.99 | 61.68 | 69.45 | FAIL |
| BigBird | 40.83 | 36.05 | 64.02 | 74.87 | FAIL |
| Linear trans. | 42.34 | 16.13 | 65.90 | 75.30 | FAIL |
| Performer | 42.77 | 18.01 | 65.40 | 77.05 | FAIL |
| S4 | **87.26** | **58.35** | **76.02** | **86.05** | **88.10** |
| RSSM-r (ours) | 49.11 | 31.50 | 64.40 | 66.46 | 54.06 |
| RSSM (ours) | 49.66 | 23.54 | 62.36 | 85.44 | 59.36 |

Its adaptability is evident in task-specific hyperparameter tuning (see Appendix H): batch sizes are adjusted to sequence length, state-space decay parameters ($m_{\mathbf{A}}$, $M_{\mathbf{A}}$) are tuned for retention vs. forgetting, and sampling rates ($\Delta$) are lowered for finer temporal resolution on longer sequences.

While S4 achieves the highest overall LRA accuracy—particularly on Path-X and ListOps—RSSM remains competitive, nearly matching S4 on Pathfinder and outperforming all transformers on Path-X. Additionally, RSSM strikes a strong balance between accuracy and efficiency, as evident from Table 3.

Unlike S4 and other fully trainable models, RSSM does not rely on common deep learning enhancements such as dropout, bidirectional layers, batch normalization, pooling, or warm restarts Loshchilov & Hutter (2016). The reported results thus reflect the raw capacity of the RSSM architecture.

## 6 Conclusions

This work introduces a novel neural architecture that combines SSMs with RC to capture long-term dependencies efficiently. Leveraging the linearity of SSMs, the model constructs parallelizable convolutional reservoirs with high memory capacity, requiring training only in a lightweight feed-forward readout. Inspired by deep RC, it produces expressive hierarchical outputs while maintaining computational efficiency. Although inheriting some limitations of RC—such as sensitivity to reservoir design and fixed internal dynamics—the integration with SSMs mitigates these challenges. Experimental results demonstrate a favorable balance between accuracy and efficiency. These benefits are crucial for real-world applications demanding high performance with limited resources. Furthermore, the model's low computational cost reduces energy use, supporting sustainability goals like the EU's 2050 climate neutrality strategy Commission (2021) and addressing concerns over deep learning's energy demands. Future work will explore alternative reservoirs, advanced optimization, and applications in forecasting, unsupervised learning, and neuromorphic hardware.

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

## A    S4D ABLATION STUDY

We describe the experiments conducted on S4D that inspired the architecture of the proposed RSSM block.

Our RSSM block does not include a mixing layer (Figure 3) that combines the $H$ features at each step of the output sequence generated by the convolutional layer. We made this architectural choice (Figure 2) because, without proper training, a mixing layer ruins the dynamics of the individual features in the output signal.

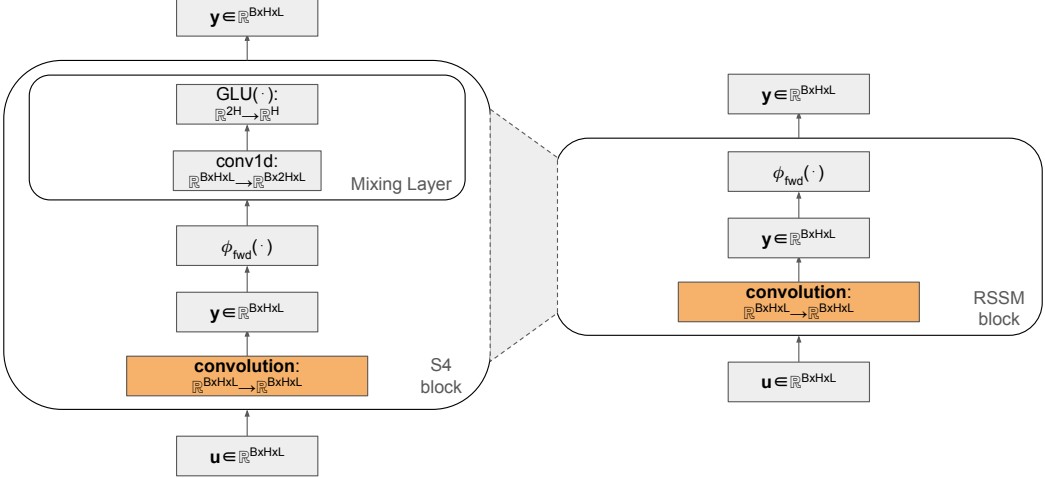

Figure 2: Graphical comparison of a single S4 block (left) and a single RSSM block (right). In the RSSM block, the mixing layer is removed to simplify the model and enhance its efficiency. Only the nonlinear activation function is retained, streamlining the architecture while maintaining essential functionality.

In our model, feature mixing is implicitly handled by the encoder through the $\mathbf{W}_{en}$ matrix, which combines the original $H_0$ features of the input signal. Learning how the features depend on each other is left to the readout model, which operates in a high-dimensional space. To demonstrate that the untrained mixing layer ruins the dynamics of individual features of the output signal, we conduct an ablation study on the S4D model to identify which parameters can be fixed without significantly affecting performance. We fix the sampling frequency $\Delta$ time and, in succession, the encoder, the parameters $\mathbf{A}$, $\mathbf{B}$, $\mathbf{C}$, and $\mathbf{D}$ of the S4D block, and the mixing layer within the S4D block. The mixing layer learns the dependencies between the features (Figure 3) Gu et al. (2022a) to overcome the fact that the S4 and S4D models are single input and single output (SISO), unlike S5, which performs a scan with an associative binary operation, in parallel Smith et al. (2023).

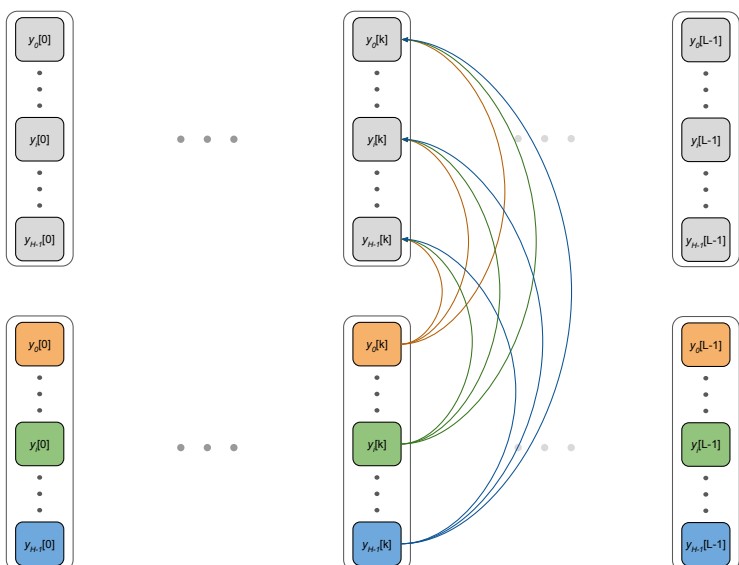

Figure 3: Graphical representation of the mixing layer. The mixing layer combines the features at each timestep independently of the other timesteps in the sequence. When trained, it can provide an advantage; otherwise, it disrupts the dynamics produced by the independent $H$ SSMs that process the features independently (Figure 4).

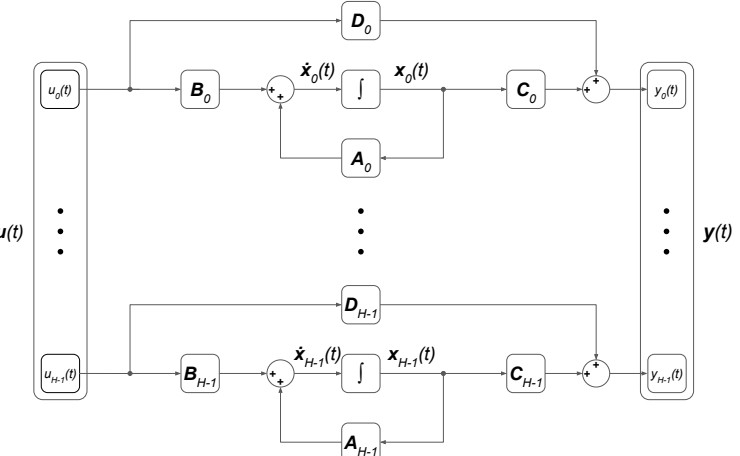

Figure 4: Graphical representation of the $H$ independent SSM, one for each feature of the input sequence. The internal state of each SSM is a vector of dimension $P$. The output signal has the same number of features $H$ as the input signal.

We notice that freezing parameters $\mathbf{B}$, $\mathbf{C}$, and $\mathbf{D}$ degrades the performance of the SSM very slightly. Freezing matrix $\mathbf{A}$ can cause performance degradation that we can avoid with a good choice of A eigenvalues.

Instead, fixing the mixing layer causes a high degradation as it ruins the trajectories over time of each feature $y(t)$ (Figure 3). Therefore, we obtain the best performance by removing the mixing layer (Table 5, and Figure 2) because the original input signal features are already mixed by the matrix $\mathbf{W}_{en}$. Moreover, the readout model can then specifically learn the feature dependencies in the output signal generated by the reservoir, independent of the temporal dependencies.

Table 5: Ablation study on the S4D model. The symbol * stands for an untrained parameter. The last three rows represent untrained S4D blocks, where only the decoder layer is trained. These experiments are run on sMNIST with a small S4D model with two layers of 64 features $H$.

| $\Delta$ | Encoder | A | B | C | D | Mixing Layer | Accuracy |
|---|---|---|---|---|---|---|---|
| * | | | | | | | 98.23 |
| * | * | | | | | | 98.63 |
| * | * | * | | | | | 98.20 |
| * | * | * | * | | | | 97.67 |
| * | * | * | * | * | | | 96.89 |
| * | * | * | * | * | * | | 97.58 |
| * | * | * | * | * | * | reservoir+glu | 73.17 |
| * | * | * | * | * | * | reservoir+tanh | 69.84 |
| * | * | * | * | * | * | **identity** | **78.75** |

## B   TRAINING PIPELINE

Our model efficiently uses both CPU and GPU processing units. Initially, input data is transferred to the GPU's dedicated memory (VRAM) to take advantage of the parallelization capabilities of convolutional operations by processing input sequences in batches on the GPU. Once the GPU completes its computations, to manage the limited memory available on GPUs VRAM, we transfer each batched global output $\mathbf{y} \in \mathbb{R}^{B \times nH \times L}$, generated by the RSSM (Equation 6), back to the system's main memory RAM. This transfer enables the CPU to collect the outputs to form a new input dataset $\mathbf{Y} \in \mathbb{R}^{N \times nH \times L}$, where $N$ represents the total number of time series (see Figure 5). This dataset, together with the original labels, is prepared for a data loader. The data loader is responsible for moving each batch (which may vary in size from $B$) back to the GPU, where a readout neural network is trained (as shown in Figures 1 and 5).

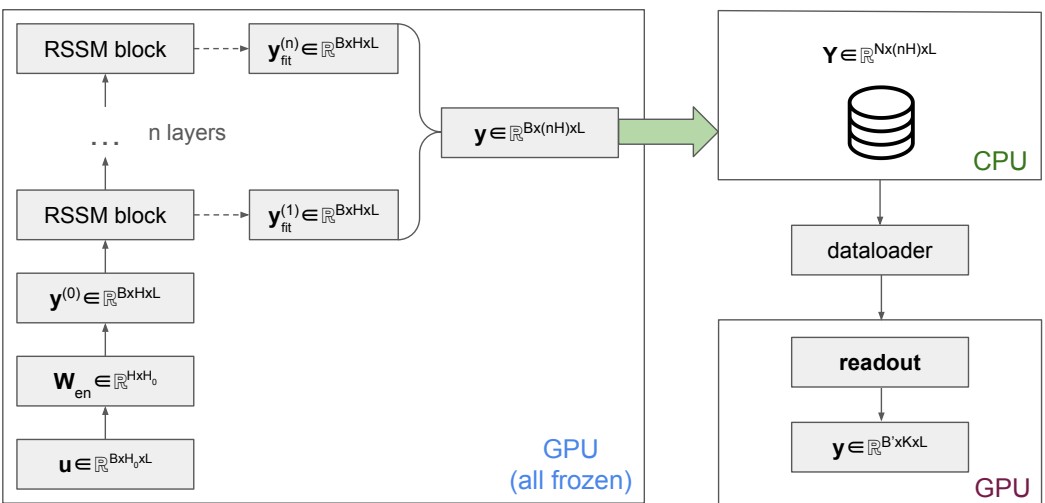

Figure 5: Graphical representation of the RSSM forward phase. Input data is transferred to the GPU's VRAM for parallel processing in the convolutional RSSM blocks. For memory efficiency, the outputs are moved to the system's RAM via the CPU, then transferred back to the GPU in batches for training the readout model. The batch size for the frozen RSSM is chosen to speed up the forward phase, while the readout model's batch size is chosen to maximize accuracy through standard model selection.

To train the readout model, the first $w$ timesteps from each time series $\mathbf{y}$ may be discarded for general tasks, as they are less representative of the series' dynamics. For the classification task, by setting $w = L - 1$, we use only the last timestep $\mathbf{y}_{L-1}$, being the most representative of the series as it encapsulates all preceding information.

The readout can be a sequence modeling neural network, such as an RNN, a 1-D convolutional network, or a trainable SSM, trained on the input dataset $\mathbf{Y} \in \mathbb{R}^{N \times nH \times (L-w)}$. Alternatively, for greater efficiency, we use a readout model that treats each timestep independently, without temporal or sequential dependencies like a closed-form readout such as ridge regression or a feed-forward neural network such as MLP. In this case, the readout is trained on the input dataset $\mathbf{Y} \in \mathbb{R}^{N(L-w) \times nH}$, among the vector of labels $\mathbf{d} \in \mathbb{R}^{N(L-w)}$.

We choose to use the MLP or the ridge regression as readout to maximize the information and expressiveness of the global RSSM output while keeping the training phase more efficient (for complexity analysis, see Section 4.4 and Table 1).

More in detail, given the vector of labels $\mathbf{d} \in \mathbb{R}^{N(L-w)}$, the ridge regression readout solves the $l$-2 regularized least-squares problem Lukoševičius & Jaeger (2009):

$$\mathbf{W}_{out} = \arg \min_{\mathbf{W}} \left\{ \|\mathbf{Y} \cdot \mathbf{W} - \mathbf{d}\|^2 + \alpha \|\mathbf{W}\|^2 \right\} \tag{13}$$

in closed form

$$\mathbf{W}_{out} = \left( \mathbf{Y}^T \mathbf{Y} + \alpha \mathbf{I} \right)^{-1} \mathbf{Y}^T \mathbf{d}. \tag{14}$$

## C    RSSM PARAMETERS

The parameters of the RSSM include the encoding matrix $\mathbf{W}_{en} \in \mathbb{R}^{H_0 \times H}$ for the encoding layer and the parameters of each RSSM block, obtained from a discretized continuous SSM.

The $\mathbf{W}_{en}$ matrix encodes the original input signal $\mathbf{u} \in \mathbb{R}^{H_0 \times L}$ in a output signal $\mathbf{y}^{(0)}$ with $H$ features. For each time step $l$, the features of $\mathbf{y}_l^{(0)}$ are given by the linear combination of the features of $\mathbf{u}_l$ with the rows of $\mathbf{W}_{en}$ as coefficients. We fix the parameter $\mathbf{W}_{en} \in \mathbb{R}^{H_0 \times H}$ such that each value has an absolute value in the interval $[m_{en}, M_{en}]$ where $m_{en}$ and $M_{en}$ are positive values. Recall that $\mathbf{W}_{en}$ have real values, so they can take values in $[-M_{en}, -m_{en}] \cup [m_{en}, M_{en}]$. Having the real values of $\mathbf{W}_{en}$ different signs helps to mix better the features of the original input signals and avoid exploding values of the encoded input $\mathbf{y}^{(0)}$ obtainable with all positive or all negative values of $\mathbf{W}_{en}$.

### C.1    RSSM BLOCK

The parameters of the RSSM block consist of the diagonal state matrix $\mathbf{A} \in \mathbb{C}^{P \times H}$, the vector of the sampling rates $\Delta \in \mathbb{R}^H$, the input matrix $\mathbf{B} \in \mathbb{C}^{P \times H}$, the output matrix $\mathbf{C} \in \mathbb{C}^{H \times P}$, and the skip connection vector $\mathbf{D} \in \mathbb{R}^H$. We recall that the parameters $\left( \mathbf{A} \in \mathbb{C}^{P \times H}, \Delta \in \mathbb{R}^H, \mathbf{B} \in \mathbb{C}^{P \times H}, \mathbf{C} \in \mathbb{C}^{H \times P}, \mathbf{D} \in \mathbb{R}^H \right)$ identify $H$ independent continuous SSMs, one for each feature $i = 0, \ldots, H-1$ as we can see in Equation 15 and Figure 4

$$\mathbf{A} = \left[ \ \mathbf{A}_0 \ | \ \cdots \ | \ \mathbf{A}_{H-1} \ \right], \ \mathbf{B} = \left[ \ \mathbf{B}_0 \ | \ \cdots \ | \ \mathbf{B}_{H-1} \ \right] \in \mathbb{C}^{P \times H}$$

$$\mathbf{C} = \begin{bmatrix} \mathbf{C}_0 \\ \hline \vdots \\ \hline \mathbf{C}_{H-1} \end{bmatrix} \in \mathbb{C}^{P \times H},$$

$$\mathbf{D} = \begin{bmatrix} \mathbf{D}_0 \\ \hline \vdots \\ \hline \mathbf{D}_{H-1} \end{bmatrix}, \ \Delta = \begin{bmatrix} \Delta_0 \\ \hline \vdots \\ \hline \Delta_{H-1} \end{bmatrix} \in \mathbb{R}^H. \tag{15}$$

The state matrix $\mathbf{A}$ has a crucial role in determining the internal dynamics of our models since it regulates the relationship between successive internal states over time. We fix the state matrix

$$\mathbf{A} = \begin{bmatrix} \lambda_{0,0} & \cdots & \lambda_{0,H-1} \\ \vdots & \vdots & \vdots \\ \lambda_{P-1,0} & \cdots & \lambda_{P-1,H-1} \end{bmatrix} \tag{16}$$

of the $H$ continuous SSMs randomly in the space

$$\left\{ \mathbf{A} \in \mathbb{C}^{P \times H} \mid m_{\mathbf{A}} \le \Re(\lambda_{i,j}) \le M_{\mathbf{A}}, \ 0 \le \Im(\lambda_{i,j}) < 2\pi \right\}. \tag{17}$$

The hyperparameters $m_{\mathbf{A}}$, $M_{\mathbf{A}}$, referring to the minimum and maximum values of the real parts of the complex eigenvalues of state matrix $\mathbf{A}$, control the internal stability of the RSSM (for more details see the Section 4.1).

The vector $\Delta \in \mathbb{R}^H$ represents the sampling rates for each input feature, used to discretize the $H$ continuous systems into $H$ discrete-time systems. We set the hyperparameters $m_\Delta$ and $M_\Delta$ that constraints the value of each $\Delta_i$:

$$\Delta \in [m_\Delta, M_\Delta]^H \subset \mathbb{R}^H, \qquad \text{with } M_\Delta \geq m_\Delta > 0. \tag{18}$$

The input matrix $\mathbf{B}$ and the output matrix $\mathbf{C}$ are crucial in defining the relationship between the state and observations. Specifically, the input matrix $\mathbf{B}$ maps the control inputs, regulating to what extent the control inputs affect the internal dynamics. On the other hand, the output matrix $\mathbf{C}$ links the internal state of the system to the observable outputs, thereby determining how the state is reflected in the measurements. We fix the parameters $\mathbf{B} \in \mathbb{C}^{P \times H}$ and $\mathbf{C} \in \mathbb{C}^{H \times P}$ of the $H$ continuous SSMs such that each value has an absolute value in the interval $[m_{\mathbf{B}}, M_{\mathbf{B}}]$ and $[m_{\mathbf{C}}, M_{\mathbf{C}}]$ respectively, where $m_{\mathbf{B}}$, $m_{\mathbf{C}}$, $M_{\mathbf{B}}$ and $M_{\mathbf{C}}$ are positive values. $\mathbf{B}$, $\mathbf{C}$ having complex values assume values belonging to the ring $\{z \in \mathbb{C} \mid m_{\mathbf{B}} \leq \|z\| \leq M_{\mathbf{B}}\}$ and $\{z \in \mathbb{C} \mid m_{\mathbf{C}} \leq \|z\| \leq M_{\mathbf{C}}\}$ respectively.

The skip connection parameter $\mathbf{D} \in \mathbb{R}^H$ operates feature-wise, adding to each output feature $i$ the respective input feature scaled by $\mathbf{D}_i$. We set its values in the interval $[m_{\mathbf{D}}, M_{\mathbf{D}}]$ where $M_{\mathbf{D}} > m_{\mathbf{D}}$ are real values. In this way, having the possibility to set the parameter $\mathbf{D}$ values to be all positive, all negative, or with mixed signs, we have more control over the skip connection.

# D   DEFINITIONS

In this appendix, we provide definitions of the main concepts used in this work. We start with the definition of a continuous-time state space model and its discretization through the Zero-Order Hold (ZOH) transform. We then define the convolution kernel associated with the discrete state space model.

**Definition D.1** (Continuous State Space Model (SSM)). A continuous SSM is a parameterized linear time-invariant system defined by

$$\begin{cases} \mathbf{x}'(t) = \mathbf{A}\mathbf{x}(t) + \mathbf{B}u(t) \\ y(t) = \Re\left(\mathbf{C}\mathbf{x}(t)\right) + \mathbf{D}u(t), \quad t \in \mathbb{R}_+. \end{cases} \tag{19}$$

where $\mathbf{x}(t) \in \mathbb{C}^P$ is the internal state, $u(t) \in \mathbb{R}$ is the input, $y(t) \in \mathbb{R}$ is the output. The model parameters are the state matrix $\mathbf{A} \in \mathbb{C}^{P \times P}$, the input matrix $\mathbf{B} \in \mathbb{C}^{P \times 1}$, the output matrix $\mathbf{C} \in \mathbb{C}^{1 \times P}$, and the skip connection $\mathbf{D} \in \mathbb{R}$.

**Definition D.2** (Discrete State Space Model (SSM)). The discrete SSM is

$$\begin{cases} \mathbf{x}_k = \overline{\mathbf{A}}\mathbf{x}_{k-1} + \overline{\mathbf{B}}u_k \\ y_k = \Re\left(\overline{\mathbf{C}}\mathbf{x}_k\right) + \overline{\mathbf{D}}u_k, \quad k = 0, \dots, L-1 \end{cases} \tag{20}$$

where the new parameters $\overline{\mathbf{A}}, \overline{\mathbf{B}}, \overline{\mathbf{C}}, \overline{\mathbf{D}}$ are obtained by a discretization method, e.g., ZOH (Definition D.3), using a constant sampling frequency $\Delta$.

**Definition D.3** (Zero-Order Hold (ZOH) Transform). Given a continuous SSM (Definition D.1) and sampling interval $\Delta > 0$, the Zero-Order Hold (ZOH) transform maps the continuous system to the discrete SSM (Definition D.2) where the discrete parameters are given by

$$\begin{cases} \overline{\mathbf{A}} = e^{\Delta \mathbf{A}} \\ \overline{\mathbf{B}} = \mathbf{A}^{-1}\left(e^{\Delta \mathbf{A}} - \mathbf{I}\right)\mathbf{B} \\ \overline{\mathbf{C}} = \mathbf{C} \\ \overline{\mathbf{D}} = \mathbf{D} \end{cases} \tag{21}$$

with $e^{\Delta \mathbf{A}}$ denoting the matrix exponential and $\mathbf{I}$ the identity matrix.

**Definition D.4** (SSM Convolution Kernel). Given a discrete SSM (Definition D.2), the convolution kernel $\overline{\mathbf{K}} \in \mathbb{C}^{1 \times L}$ is defined as

$$\overline{\mathbf{K}} = \left[ \overline{\mathbf{CB}}, \dots, \overline{\mathbf{CA}}^l \overline{\mathbf{B}}, \dots, \overline{\mathbf{CA}}^{L-1} \overline{\mathbf{B}} \right] \tag{22}$$

where $L$ is the sequence length. The kernel $\overline{\mathbf{K}}$ defines the linear convolution mapping from the input sequence $u_k$ to the output sequence $y_k$, obtained by unrolling the recurrence of the discrete SSM (Definition D.2):

$$y_k = \Re \left( \sum_{l=0}^{k} \overline{\mathbf{K}}_l \, u_{k-l} \right) + \overline{\mathbf{D}} \, u_k. \tag{23}$$

**Definition D.5** (Convolution Kernel for Diagonal Discrete SSM). Let $\overline{\mathbf{A}} = \text{diag}(\lambda_0, \dots, \lambda_{P-1}) \in \mathbb{C}^{P \times P}$ be a diagonal state matrix, $\overline{\mathbf{B}} \in \mathbb{C}^{P \times 1}$ the input matrix, and $\overline{\mathbf{C}} \in \mathbb{C}^{1 \times P}$ the output matrix. The convolution kernel $\overline{\mathbf{K}} \in \mathbb{C}^{1 \times L}$ in Definition D.4 can be written as:

$$\overline{\mathbf{K}} = \left( \overline{\mathbf{C}} \odot \overline{\mathbf{B}}^T \right) \cdot \mathbf{V}, \tag{24}$$

where $\odot$ denotes element-wise multiplication and $\mathbf{V} \in \mathbb{C}^{P \times L}$ is the Vandermonde matrix generated by $(\lambda_0, \dots, \lambda_{P-1})$, with entries $\mathbf{V}_{i,l} = \lambda_i^l$

**Definition D.6** (Convolution View of SSM). Given a discrete SSM (Definition D.2), the kernel $\overline{\mathbf{K}}$ (Definition D.4) defines the linear convolution mapping from the input sequence $\mathbf{u} \in \mathbb{R}^{1 \times L}$ to the output sequence $\mathbf{y} \in \mathbb{R}^{1 \times L}$

$$\mathbf{y} = \Re \left( \overline{\mathbf{K}} * \mathbf{u} \right) + \overline{\mathbf{D}} \, \mathbf{u}. \tag{25}$$

The linear convolution is computed using the discrete Fourier transform (DFT):

$$\mathbf{y} = \Re \left( DFT^{-1} \left( DFT \left( \overline{\mathbf{K}} \right) \odot DFT \left( \mathbf{u} \right) \right) \right) + \overline{\mathbf{D}} \, \mathbf{u}. \tag{26}$$

# E   PROOFS

This appendix provides proofs for the theorems and lemmas presented in the main text. For all the proofs, we assume the number of features $H = 1$ for simplicity. This assumption does not affect the generality of the results, as each feature is processed independently in the RSSM layer.

**Theorem 4.1: Stability Conditions**

*Proof.* First, we note that a hierarchy of contractive reservoirs is stable Gallicchio & Micheli (2017). Second, we note that for a diagonal state space matrix, the eigenvalues coincide with the singular values. Therefore, the spectral norm coincides with the spectral radius. Hence, if the spectral radius is less than 1, then the corresponding dynamics are contractive. Moreover, an RSSM block is composed of $H$ independent SSM systems. With these remarks, it now suffices to find conditions of stability for a single SSM to imply the stability of a deep RSSM architecture. Let $\mathbf{A} = \text{diag}(\lambda_0, \dots, \lambda_P) \in \mathbb{C}^{P \times P}$ be a diagonal continuous-time state matrix. The continuous-time SSM is stable if and only if the real part of every eigenvalue is non-positive, i.e., $\Re(\lambda_i) \leq 0$ for all $i$ (see, e.g., Khalil (2002); Ogata (2010); Hespanha (2018)).

Consider the discretization of the system with sampling interval $\Delta > 0$. The discretized state matrix is given by

$$\overline{\mathbf{A}} = \exp(\Delta \mathbf{A}) = \text{diag}(e^{\Delta \lambda_0}, \dots, e^{\Delta \lambda_P}).$$

Let $\overline{\lambda}_i = e^{\Delta \lambda_i}$ denote the $i$-th eigenvalue of $\overline{\mathbf{A}}$. The modulus of $\overline{\lambda}_i$ is

$$\rho_i = |\overline{\lambda}_i| = |e^{\Delta \lambda_i}| = e^{\Delta \Re(\lambda_i)}.$$

The discrete-time SSM is stable if and only if $|\overline{\lambda}_i| \leq 1$ for all $i$ (see, e.g., Kailath (1980); Ogata (1997)). This is equivalent to

$$e^{\Delta \Re(\lambda_i)} \leq 1 \quad \Longleftrightarrow \quad \Delta \Re(\lambda_i) \leq 0.$$

Since $\Delta > 0$, this holds if and only if $\Re(\lambda_i) \leq 0$ for all $i$.

Therefore, the stability condition for the continuous SSM ($\Re(\lambda_i) \leq 0$) is equivalent to the stability condition for the discretized SSM ($\rho_i \leq 1$). $\qquad \square$

**Lemma 4.1: State Dimension Effect**

*Proof.* The pre-activated output of the RSSM layer is given by

$$\mathbf{y} = \Re\left(DFT^{-1}\left(DFT\left(\overline{\mathbf{K}}\right) \odot DFT\left(\mathbf{u}\right)\right)\right) + \overline{\mathbf{D}}\mathbf{u}, \tag{27}$$

where $DFT$ is the discrete Fourier transform and $DFT^{-1}$ is its inverse. Assuming the skip connection $\overline{\mathbf{D}} = 0$, we can rewrite the Equation 27 as

$$\mathbf{y} = \Re(\overline{\mathbf{K}} * \mathbf{u}) = \Re(\overline{\mathbf{K}}) * \mathbf{u} = \overline{\mathbf{F}} * \mathbf{u},$$

where we denote the convolution filter as $\overline{\mathbf{F}} = \Re(\overline{\mathbf{K}})$. The kernel $\overline{\mathbf{K}}$ is given by $\overline{\mathbf{K}} = \left(\overline{\mathbf{C}} \odot \overline{\mathbf{B}}^T\right) \cdot \mathbf{V}$, where $\mathbf{V}$ is the Vandermonde matrix of $\overline{\mathbf{A}}$. To simplify, we define the vector $\mathbf{W} = \overline{\mathbf{C}} \odot \overline{\mathbf{B}}^T$. The $l$-th element of the kernel $\overline{\mathbf{K}}$ is given by the dot product $\mathbf{W} \cdot \overline{\mathbf{A}}^l$. Rewriting $\mathbf{W}$ and $\overline{\mathbf{A}}^l$ in polar notation:

$$\mathbf{W} = \overline{\mathbf{C}} \odot \overline{\mathbf{B}}^T = \begin{bmatrix} \nu_0 e^{i\psi_0} \cdots \nu_P e^{i\psi_P} \end{bmatrix}$$

and

$$\overline{\mathbf{A}}^l = \begin{bmatrix} \rho_0^l e^{i(l\theta_0)} \\ \vdots \\ \rho_P^l e^{i(l\theta_P)} \end{bmatrix},$$

the $l$-th element of the kernel becomes $\overline{\mathbf{K}}_l = \sum_{i=0}^{P-1} \nu_i \rho_i^l \cdot e^{i(\psi_i + l\theta_i)}$. Thus, the $l$-th element of the convolution filter is:

$$\overline{\mathbf{F}}_l = \sum_{i=0}^{P-1} \nu_i \rho_i^l \cdot \cos(\psi_i + l\theta_i).$$

The convolution filter $\overline{\mathbf{F}}$ is equivalent to a sum of $P$ oscillatory components, where $P$ is the state dimension of the SSM:

$$\overline{\mathbf{F}} = \sum_{i=0}^{P-1} \overline{\mathbf{F}}^{(i)}.$$

Therefore, the $i$-th oscillatory component is $\overline{\mathbf{F}}^{(i)} \in \mathbb{R}^{1 \times L}$, with the $l$-th element $\overline{\mathbf{F}}_l^{(i)} = \nu_i \rho_i^l \cos(\psi_i + l\theta_i)$. Each component $\overline{\mathbf{F}}^{(i)}$ is characterized by its amplitude $\nu_i$, decay rate $\rho_i$, phase $\psi_i$, and frequency $\theta_i$. $\qquad\square$

**Lemma 4.2: Stability Constraint**

*Proof.* Recall that the $l$-th element of the convolution filter is given by

$$\overline{\mathbf{F}}_l = \sum_{i=0}^{P-1} \nu_i \rho_i^l \cos(\psi_i + l\theta_i),$$

where $\nu_i \geq 0$ is the amplitude, $\rho_i = e^{\Delta\Re(\lambda_i)}$ is the decay/growth factor, and $\psi_i, \theta_i$ are the phase and frequency parameters.

Since $|\cos(\psi_i + l\theta_i)| \leq 1$ for all $i$ and $l$, we have

$$|\overline{\mathbf{F}}_l| \leq \sum_{i=0}^{P-1} \nu_i |\rho_i|^l.$$

Moreover, since $\rho_i > 0$ for all $i$, we can bound the sum as

$$\sum_{i=0}^{P-1} \nu_i \min_i \{\rho_i\}^l \leq \sum_{i=0}^{P-1} \nu_i \rho_i^l \leq \sum_{i=0}^{P-1} \nu_i \max_i \{\rho_i\}^l.$$

Thus,

$$\left(\sum_{i=0}^{P-1} \nu_i\right) \min_i\{\rho_i\}^l \leq |\overline{\mathbf{F}}_l| \leq \left(\sum_{i=0}^{P-1} \nu_i\right) \max_i\{\rho_i\}^l.$$

To ensure stability (i.e., boundedness of the filter for all $l$), it is necessary that $\max_i\{\rho_i\} \leq 1$, which is equivalent to requiring $\Re(\lambda_i) \leq 0$ for all $i$. If $\Re(\lambda_i) < 0$, then $\rho_i < 1$ and the filter coefficients decay exponentially, resulting in fading memory. If $\Re(\lambda_i) = 0$, then $\rho_i = 1$ and the filter coefficients remain bounded but do not decay.

Therefore, the real part of the eigenvalues controls both the stability and the memory capacity of the convolution filter, as claimed. $\square$

### Lemma 4.3: Parameter Scaling

*Proof.* Let $\mathbf{B} \in \mathbb{C}^{P \times 1}$ and $\mathbf{C} \in \mathbb{C}^{1 \times P}$ be the input and output matrices of the continuous SSM, with each entry of $\mathbf{B}$ and $\mathbf{C}$ having modulus in $[m_{\mathbf{B}}, M_{\mathbf{B}}]$ and $[m_{\mathbf{C}}, M_{\mathbf{C}}]$, respectively, where $m_{\mathbf{B}}, m_{\mathbf{C}}, M_{\mathbf{B}}, M_{\mathbf{C}} > 0$ (see Appendix C).

After discretization (e.g., via the ZOH method), the discrete parameters are

$$\overline{\mathbf{A}} = e^{\Delta\mathbf{A}}, \quad \overline{\mathbf{B}} = \mathbf{A}^{-1}(e^{\Delta\mathbf{A}} - \mathbf{I})\mathbf{B}, \quad \overline{\mathbf{C}} = \mathbf{C}.$$

The convolution kernel is given by

$$\overline{\mathbf{K}} = (\overline{\mathbf{C}} \odot \overline{\mathbf{B}}^T) \cdot \mathbf{V},$$

where $\mathbf{V}$ is the Vandermonde matrix of $\overline{\mathbf{A}}$. Define $\mathbf{W} = \overline{\mathbf{C}} \odot \overline{\mathbf{B}}^T$.

Expressing $\mathbf{W}$ in polar form, each entry is $\mathbf{W}_i = \nu_i e^{i\psi_i}$, where $\nu_i = |\overline{\mathbf{B}}_i||\overline{\mathbf{C}}_i|$ and $\psi_i = \arg(\overline{\mathbf{B}}_i\overline{\mathbf{C}}_i) = \arg(\overline{\mathbf{B}}_i) + \arg(\overline{\mathbf{C}}_i)$. Since $|\overline{\mathbf{B}}_i| \propto |\mathbf{B}_i| \in [m_{\mathbf{B}}, M_{\mathbf{B}}]$ and $|\overline{\mathbf{C}}_i| = |\mathbf{C}_i| \in [m_{\mathbf{C}}, M_{\mathbf{C}}]$, it follows that $[m_{\mathbf{B}} m_{\mathbf{C}}, M_{\mathbf{B}} M_{\mathbf{C}}]$ it give us an estimate of the amplitude $\nu_i$. The phase $\psi_i$ is in $[0, 2\pi)$ due to the randomness of $\arg(\mathbf{B}_i)$, and $\arg(\mathbf{C}_i)$ (see Appendix C).

Thus, the amplitude and phase of each oscillatory component in the convolution filter are directly determined by the magnitudes and arguments of the entries of $\overline{\mathbf{B}}$ and $\overline{\mathbf{C}}$, as claimed. $\square$

### Lemma 4.4: Sampling Rate Effect

*Proof.* The Zero-Order Hold (ZOH) method discretizes the continuous-time SSM by sampling the input signal at intervals of $\Delta$, i.e., $u_k = u(k\Delta)$, and assumes the signal is piecewise constant between samples:

$$u((k+\delta)\Delta) = u(k\Delta), \quad \delta \in [0, 1).$$

Under ZOH, the continuous SSM parameters $(\mathbf{A}, \mathbf{B}, \mathbf{C}, \mathbf{D})$ are mapped to discrete parameters as follows:

$$\begin{cases} \overline{\mathbf{A}} = e^{\Delta\mathbf{A}} \\ \overline{\mathbf{B}} = \mathbf{A}^{-1}\left(e^{\Delta\mathbf{A}} - \mathbf{I}\right)\mathbf{B} \\ \overline{\mathbf{C}} = \mathbf{C} \\ \overline{\mathbf{D}} = \mathbf{D} \end{cases}$$

Let $\lambda_i = \Re(\lambda_i) + i\Im(\lambda_i)$ be the $i$-th eigenvalue of $\mathbf{A}$. The $i$-th eigenvalue of $\overline{\mathbf{A}}$ is $\overline{\lambda}_i = e^{\Delta\lambda_i} = e^{\Delta\Re(\lambda_i)}e^{i\Delta\Im(\lambda_i)}$. Thus, the modulus and argument are

$$\rho_i = |\overline{\lambda}_i| = e^{\Delta\Re(\lambda_i)}, \qquad \theta_i = \arg(\overline{\lambda}_i) = \Delta\Im(\lambda_i).$$

The convolution filter is given by

$$\overline{\mathbf{F}}_l = \sum_{i=0}^{P-1} \nu_i \rho_i^l \cos(\psi_i + l\theta_i),$$

where $\nu_i$ and $\psi_i$ are determined by $\overline{\mathbf{B}}$ and $\overline{\mathbf{C}}$.

As $\Delta$ increases, $\theta_i$ increases linearly, resulting in a higher oscillation frequency in the filter. Simultaneously, for $\Re(\lambda_i) < 0$, increasing $\Delta$ decreases $\rho_i$ (since $0 < \rho_i < 1$), causing the filter coefficients to decay more rapidly and thus reducing memory capacity. Conversely, decreasing $\Delta$ yields smoother filters (lower frequency) and increases memory capacity (slower decay).

Therefore, the sampling rate $\Delta$ directly controls both the frequency and decay rate of each oscillatory component in the convolution filter, as claimed. $\qquad\square$

## F  EXPERIMENTAL SETUP

In this appendix, we describe the architecture of our model for classification tasks and outline the models used as benchmarks.

To exploit the memory capacity of the RSSM and improve training efficiency, we employ two efficient readouts: an MLP and a ridge regressor, with the latter trained in closed form (Equations **??**, 14).

The model using the MLP readout is referred to as RSSM, while we denote the ridge regressor variant as RSSM-r. The readout learns dependencies between features within each timestep, independently of other timesteps, while the reservoir component of the RSSM captures temporal dependencies in the input sequences without requiring training.

Only the final timestep $\mathbf{y}_{L-1}$ is used for classification tasks, as it encapsulates all previous states and best represents the entire sequence. The input dimension for the ridge regressor and the first MLP layer corresponds to the global output dimension of the RSSM, $n \cdot H$, where $n$ is the number of RSSM layers, and $H$ is the number of output sequence features per layer (see Section 3.2 and Figure 1). Each MLP layer uses the Gated Linear Unit (GLU) activation function, defined as:

$$GLU(\cdot) : \mathbb{R}^{2N} \longrightarrow \mathbb{R}^N$$
$$\begin{bmatrix} \mathbf{a} \\ \mathbf{b} \end{bmatrix} \longrightarrow \mathbf{a} \odot \sigma(\mathbf{b}), \tag{28}$$

where $\odot$ denotes element-wise multiplication and $\sigma(\cdot)$ is the sigmoid function. The GLU activation employs a gating mechanism that helps select important features, effectively halving the output dimensionality at each layer and controlling information flow, thus reducing readout complexity while retaining the most relevant features Dauphin et al. (2016).

We evaluate our model's performance against two fully trainable models, the GRU and S4, and two reservoir computing models: the Leaky Integrator ESN Jaeger et al. (2007) in its shallow configuration (ESN) and deep configuration (DeepESN) Gallicchio et al. (2017).

The ESN consists of a single layer, with the number of neurons matching the total number of hidden features in the RSSM with $n$ layers. We configure the DeepESN with the same number of layers and neurons per layer. The states outputted from each layer of the DeepESN are concatenated, similar to our architecture (see Figure 1) Gallicchio et al. (2017). Both ESN models use a closed-form ridge classifier for the readout, as in RSSM-r.

The GRU and S4 models are configured with comparable numbers of layers and hidden units to ensure fair efficiency comparisons. For reservoir-based models (ESN, RSSM-r, and RSSM), we set the batch size $B$ as large as possible to optimize GPU efficiency, as the forward pass is independent of batch size. For training the MLP readout, we use a separate batch size $B'$, selected through model selection, similar to the approach used for fully trainable models (GRU and S4).

During the training phase of our MLP readout model and other online training models, we split the development set into training and validation sets, using early stopping with a patience hyperparameter to prevent overfitting. After early stopping, we further train the model on the whole development set (training and validation combined) for a single epoch. We use the AdamW optimizer, which includes weight decay as a regularization hyperparameter. Additionally, a simple scheduler reduces the learning rate by a factor if the validation loss does not decrease after reaching half the patience limit. Cross-entropy loss is used as the loss function.

### F.1 MODEL SELECTION

This section details the approach used for hyperparameter tuning and selecting optimal model configurations.

For reservoir-based models (RSSM, RSSM-r, ESN, and DeepESN), which have untrained components, model selection is critical due to their sensitivity to hyperparameters and the large search space required. To address this, we employ random search, sampling up to 1000 configurations from the possible hyperparameter settings generated by a grid search. Unlike grid search, which systematically evaluates predefined combinations, random search allows for more efficient exploration of a broader range of configurations while reducing computational cost Bergstra & Bengio (2012).

In contrast, for fully trainable models (GRU, S4), which are less sensitive to hyperparameter tuning, we use a traditional grid search approach.

Appendix G provides detailed tables outlining the hyperparameter space for each model.

To improve the efficiency of model selection for the RSSM model with ridge regression and MLP readouts, we leverage the modularity of the reservoir component and the independent processing of each of the $H$ features. We implement a two-level model selection process to fully exploit these characteristics.

In the first phase, we select the best hyperparameters of the reservoir components of the RSSM by keeping the same ridge regression as readout (Table 9). We can fix the number of hidden features to a small enough value to enable a large batch size of $B$ that does not change the result of the input processing due to the untrained reservoir. The combination of reduced model complexity (from the small $H$), large batch size, and the speed of the ridge classifier significantly accelerates the reservoir hyperparameter tuning. After identifying the optimal hyperparameters, we scale the reservoir component by increasing $H$ to the desired size, and we generate the reservoir output, which serves as the readout dataset for the second phase.

In the second phase, we perform separate model selections for each readout – MLP (Table 11) and ridge regression (Table 10) – using the dataset from the selected reservoir. This allows us to fix the reservoir hyperparameters and focus on tuning the readouts.

By leveraging the independence of the $H$ features arising from the $H$ independent SSMs, the reservoir component scales with $H$. Additionally, the modularity of the reservoir component allows us to keep the same reservoir output for different readout models. These properties help mitigate the complexity of the hyperparameter space, making this two-level random search process both efficient and scalable.

Appendix H lists the final models for each task, along with the best hyperparameters obtained through the model selection. Specifically, Tables 16 and 17 show the best hyperparameters selected for pixel-by-pixel and LRA tasks, respectively, for both RSSM and RSSM-r models.

### F.2 EVALUATION METRICS

In our experiments, we evaluate the accuracy of the time-series classification tasks and assess the model's computational impact. Precisely, we measure the training time, carbon emissions, and energy consumption to comprehensively evaluate the environmental and resource costs associated with our approach. To achieve this, we use the CodeCarbon library Courty et al. (2024). The metrics collected include training time (measured in seconds), $CO_2$ emissions (measured in grams), and energy consumption (measured in kilowatt-hours, kWh). We conduct all experiments using the Tesla V100-PCIE-16GB GPU. By incorporating both accuracy and resource consumption metrics, our evaluation provides insight into the model's predictive performance and the associated environmental costs, highlighting the trade-offs between computational demands and sustainability.

## G HYPERPARAMETER SPACE

In this section, we provide details of the model selection, specifying the hyperparameter space for each model independently of the task. Tables 6 and 7 list the hyperparameters and their respective possible values for GRU and S4, respectively. Table 8 presents the hyperparameters and their possible values for both ESN and DeepESN.

Table 9 outlines the hyperparameters and their corresponding values for the reservoir component of RSSM. Lastly, Tables 10 and 11 list the hyperparameters and possible values for the ridge and MLP readouts of RSSM, respectively.

Table 6: GRU hyperparameter space for sMNIST, pMNIST, sCIFAR-10 tasks. "lr" and "wd" are, respectively, the learning rate and weight decay of AdamW. "rop" is the reducing factor of the learning rate on a validation score plateau.

| GRU | hyperparameter values |
|---|---|
| batch size | $\{64, 128\}$ |
| lr | $\{0.0005, 0.001, 0.005, 0.01\}$ |
| wd | $\{0.01, 0.05, 0.1, 0.5\}$ |
| rop | 0.2 |
| patience | 10 |

Table 7: S4 hyperparameter space for sMNIST, pMNIST and sCIFAR-10 tasks. "init" denotes the kernel structure. "num ssm" is the number of distinct SSM respect the total $H$ independent SSM. "lr" and "wd" are, respectively, the learning rate and weight decay of AdamW. "rop" is the reducing factor of the learning rate on a validation score plateau. Further hyperparameter details are in Gu et al. (2022a).

| S4 | hyperparameter values |
|---|---|
| batch size | $\{64, 128\}$ |
| state dim $P$ | $\{64, 128, 256\}$ |
| dropout | $\{0.0, 0.1, 0.2\}$ |
| tie dropout | $\{False, True\}$ |
| init | legs |
| $m_\Delta$ | 0.001 |
| $M_\Delta$ | 0.1 |
| kernel lr | 0.001 |
| kernel wd | 0.0 |
| bidirectional | $\{False, True\}$ |
| final act | $GLU(\cdot)$ |
| num ssm | $\{1, 2, 8, 64\}$ |
| lr | $\{0.0005, 0.001, 0.005, 0.01\}$ |
| wd | $\{0.01, 0.05, 0.1, 0.5\}$ |
| rop | 0.2 |
| patience | 10 |

Table 8: ESN and DeepESN hyperparameter space for sMNIST, pMNIST and sCIFAR-10 tasks. "input scaling" is the scaling factor of the input weight matrix, "rho" is the spectral radius of the recurrent weight matrix, and "leaky" is the leaking rate Nakajima & Fischer (2021); Gallicchio et al. (2017). "ridge $\alpha$" is the ridge parameter of Thikonov regularization for the offline training (Equation **??**).

| ESN/DeepESN | hyperparameter values |
|---|---|
| batch size $B$ | large as possible |
| input scaling | $\{0.1, 0.2, 0.3, 0.4, 0.5, 0.6, 0.7, 0.8, 1.0\}$ |
| bias scaling | $\{0.0, 0.1, 0.2, 0.3, 0.4, 0.5, 0.6, 0.7, 0.8, 1.0\}$ |
| rho | $\{0.8, 0.9, 1.0, 1.1\}$ |
| leaky | $\{0.3, 0.4, 0.5, 0.6, 0.7, 0.8, 0.9, 1.0\}$ |
| ridge $\alpha$ | $\{0.0, 0.4, 0.8, 1.5, 3.0, 5.0, 7.5, 10.0, 12.5, 15.0\}$ |

Table 9: Hyperparameter space of the RSSM reservoir component for sMNIST, pMNIST, sCIFAR-10, and LRA tasks. The hyperparameters of the RSSM reservoir component are described in C.1. "ridge $\alpha$" is the ridge parameter of Thikonov regularization for the offline training (Equation **??**). We fix the ridge regressor readout while selecting the RSSM reservoir component with small $H = 64$ as described in F.1.

| RSSM | hyperparameter values |
|------|------------------------|
| features $H$ | 64 |
| batch size $B$ | large as possible |
| state dim $P$ | $\{8, 16, 32, 64, 128, 256, 512, 1024\}$ |
| $\phi_{fwd}(\cdot)$ | $ReLU(\cdot)$ |
| $\phi_{fit}(\cdot)$ | $TanH(\cdot)$ |
| $m_{\mathbf{W}_{en}}$ | 0.0 |
| $M_{\mathbf{W}_{en}}$ | $\{0.1, 0.25, 0.5, 0.75, 1.0, 1.25\}$ |
| discrete | False |
| $m_{\mathbf{A}}$ | $\{-4.5, -4.0, -3.5, -3.0, -2.5, -2.0, -1.5, -1.0, -0.5, 0.0\}$ |
| $M_{\mathbf{A}}$ | $\{-0.5, 0.0\}$ |
| $m_{\Delta}$ | $\{0.0001, 0.0005, 0.001, 0.005, 0.01\}$ |
| $M_{\Delta}$ | $\{0.001, 0.005, 0.01, 0.005, 0.1\}$ |
| $m_{\mathbf{B}}$ | 0.0 |
| $M_{\mathbf{B}}$ | $\{0.1, 0.25, 0.5, 0.75, 1.0, 1.25\}$ |
| $m_{\mathbf{C}}$ | 0.0 |
| $M_{\mathbf{C}}$ | $\{0.1, 0.25, 0.5, 0.75, 1.0, 1.25\}$ |
| $m_{\mathbf{D}}$ | $\{0.0, 0.25, 0.5, 0.75, 1.0\}$ |
| $M_{\mathbf{D}}$ | $\{0.0, 0.25, 0.5, 0.75, 1.0\}$ |
| ridge $\alpha$ | 0.8 |

Table 10: Ridge regressor hyperparameter space for sMNIST, pMNIST, sCIFAR-10, and LRA tasks. $\alpha$ is the ridge parameter of Thikonov regularization for the offline training (Equation **??**).

| ridge regression | hyperparameter values |
|------------------|------------------------|
| $\alpha$ | $\{0.0, 0.4, 0.8, 1.5, 3.0, 5.0, 7.5, 10.0, 12.5, 15.0\}$ |

Table 11: MLP hyperparameter space for sMNIST, pMNIST, sCIFAR-10, and LRA tasks. For sMNIST and pMNIST, the number of layers is fixed at 2 due to the lower complexity of the task. "lr" and "we" are, respectively, the learning rate and weight decay of AdamW. "rop" is the reducing factor of the learning rate on a validation score plateau.

| MLP | hyperparameter values |
|-----|------------------------|
| layers | $\{2, 4, 6\}$ |
| batch size $B'$ | $\{64, 128\}$ |
| lr | $\{0.0005, 0.001, 0.005, 0.01\}$ |
| wd | $\{0.01, 0.05, 0.1, 0.5\}$ |
| rop | 0.2 |
| patience | 10 |

# H  BEST HYPERPARAMETERS

In this section, we present the best models selected through the model selection process for each model and task (Section F.1 and Appendix G).

Tables 12, 13, 14, 15, and 16 show the optimal hyperparameter values for GRU, S4, ESN, DeepESN, and RSSM models, respectively, across the sMNIST, pMNIST, and sCIFAR-10 tasks.

Table 17 presents the optimal hyperparameter values for the RSSM models on the LRA tasks.

Table 12: GRU best hyperparameters for sMNIST, pMNIST, and sCIFAR-10 tasks selected from Table 6.

| GRU | sMNIST | pMNIST | sCIFAR-10 |
|---|---|---|---|
| layers | 2 | 2 | 6 |
| features $P$ | 256 | 256 | 512 |
| batch | 128 | 128 | 64 |
| lr | 0.001 | 0.001 | 0.001 |
| wd | 0.1 | 0.1 | 0.1 |
| rop | 0.2 | 0.2 | 0.2 |
| patience | 10 | 10 | 10 |

Table 13: S4 best hyperparameters for sMNIST and pMNIST, and sCIFAR-10 tasks selected from Table 7.

| S4 | sMNIST | pMNIST | sCIFAR-10 |
|---|---|---|---|
| layers | 2 | 2 | 6 |
| features $H$ | 256 | 256 | 512 |
| batch | 128 | 128 | 64 |
| state dim $P$ | 64 | 64 | 64 |
| dropout | 0.1 | 0.1 | 0.1 |
| tie dropout | True | True | True |
| init | legs | legs | legs |
| $m_\Delta$ | 0.001 | 0.001 | 0.001 |
| $M_\Delta$ | 0.1 | 0.1 | 0.1 |
| kernel lr | 0.001 | 0.001 | 0.001 |
| kernel wd | 0.0 | 0.0 | 0.0 |
| bidirectional | True | True | True |
| final act | $GLU(\cdot)$ | $GLU(\cdot)$ | $GLU(\cdot)$ |
| num ssm | 1 | 1 | 2 |
| lr | 0.01 | 0.01 | 0.01 |
| wd | 0.05 | 0.05 | 0.05 |
| rop | 0.2 | 0.2 | 0.2 |
| patience | 10 | 10 | 10 |

Table 14: ESN best hyperparameters for sMNIST, pMNIST, and sCIFAR-10 tasks selected from Table 8.

| ESN | sMNIST | pMNIST | sCIFAR-10 |
|---|---|---|---|
| features $H \equiv P$ | 2048 | 2048 | 16384 |
| batch $B$ | 256 | 256 | 32 |
| input scaling | 0.3 | 0.1 | 0.5 |
| bias scaling | 0.0 | 0.0 | 0.7 |
| rho | 1.1 | 1.0 | 1.0 |
| leaky | 0.5 | 1.0 | 0.1 |
| ridge $\alpha$ | 0.4 | 0.4 | 12.5 |

Table 15: DeepESN best hyperparameters for sMNIST, pMNIST, and sCIFAR-10 tasks selected from Table 8.

| DeepESN | sMNIST | pMNIST | sCIFAR-10 |
|---|---|---|---|
| layers | 2 | 2 | 8 |
| features $H \equiv P$ | 1024 | 1024 | 2048 |
| batch $B$ | 256 | 256 | 32 |
| input scaling | 0.3 | 0.1 | 0.6 |
| bias scaling | 0.1 | 0.0 | 0.4 |
| rho | 1.0 | 1.0 | 1.0 |
| leaky | 0.2 | 1.0 | 0.7 |
| ridge $\alpha$ | 0.4 | 0.4 | 15.0 |

Table 16: RSSM-r and RSSM best hyperparameters for sMNIST, pMNIST, and sCIFAR-10 tasks. The first block of hyperparameters corresponds to the reservoir component, the second to the ridge readout, and the third to the MLP readout. The optimal hyperparameters for the reservoir, ridge readout, and MLP readout are selected from Tables 9, 10, and 11, respectively.

| RSSM | sMNIST | pMNIST | sCIFAR-10 |
|---|---|---|---|
| layers $n$ | 2 | 2 | 8 |
| features $H$ | 1024 | 1024 | 2048 |
| batch $B$ | 256 | 256 | 32 |
| state dim $P$ | 512 | 256 | 64 |
| $\phi_{fwd}(\cdot)$ | $ReLU(\cdot)$ | $ReLU(\cdot)$ | $ReLU(\cdot)$ |
| $\phi_{fit}(\cdot)$ | $TanH(\cdot)$ | $TanH(\cdot)$ | $TanH(\cdot)$ |
| $m_{\mathbf{W}_{en}}$ | 0.0 | 0.0 | 0.0 |
| $M_{\mathbf{W}_{en}}$ | 0.25 | 0.75 | 0.75 |
| discrete | False | False | False |
| $m_{\mathbf{A}}$ | $-3.5$ | $-2.5$ | 0.0 |
| $M_{\mathbf{A}}$ | 0.0 | 0.0 | 0.0 |
| $m_{\Delta}$ | 0.0001 | 0.0005 | 0.0001 |
| $M_{\Delta}$ | 0.1 | 0.1 | 0.1 |
| $m_{\mathbf{B}}$ | 0.0 | 0.0 | 0.0 |
| $M_{\mathbf{B}}$ | 1.25 | 0.75 | 0.75 |
| $M_{\mathbf{C}}$ | 1.0 | 0.75 | 0.1 |
| $m_{\mathbf{D}}$ | 0.25 | 0.0 | 0.25 |
| $M_{\mathbf{D}}$ | 0.75 | 0.75 | 1.0 |
| ridge $\alpha$ | 0.0 | 0.0 | 12.5 |
| MLP layers | 2 | 2 | 6 |
| batch $B'$ | 128 | 128 | 64 |
| lr | 0.005 | 0.005 | 0.0005 |
| wd | 0.1 | 0.1 | 0.5 |
| rop | 0.2 | 0.2 | 0.2 |
| patience | 10 | 10 | 10 |

Table 17: RSSM-r and RSSM best hyperparameters for LRA tasks. The first block of hyperparameters corresponds to the reservoir component, the second to the ridge readout, and the third to the MLP readout. The optimal hyperparameters for the reservoir, ridge readout, and MLP readout are selected from Tables 9, 10, and 11, respectively.

| RSSM | Image | ListOps | Text | Pathfinder | Path-X |
|---|---|---|---|---|---|
| layers $n$ | 8 | 8 | 8 | 8 | 8 |
| features $H$ | 2048 | 2048 | 2048 | 2048 | 2048 |
| batch $B$ | 32 | 32 | 8 | 32 | 4 |
| state dim $P$ | 1024 | 512 | 256 | 64 | 128 |
| $\phi_{fwd}(\cdot)$ | $ReLU(\cdot)$ | $ReLU(\cdot)$ | $ReLU(\cdot)$ | $ReLU(\cdot)$ | $ReLU(\cdot)$ |
| $\phi_{fit}(\cdot)$ | $TanH(\cdot)$ | $TanH(\cdot)$ | $TanH(\cdot)$ | $TanH(\cdot)$ | $TanH(\cdot)$ |
| $m_{\mathbf{W}_{en}}$ | 0.0 | 0.0 | 0.0 | 0.0 | 0.0 |
| $M_{\mathbf{W}_{en}}$ | 0.25 | 0.25 | 1.25 | 1.0 | 0.5 |
| discrete | False | False | False | False | False |
| $m_{\mathbf{A}}$ | $-2.5$ | $-0.5$ | $-1.0$ | 0.0 | $-1.5$ |
| $M_{\mathbf{A}}$ | 0.0 | $-0.5$ | $-0.5$ | 0.0 | 0.0 |
| $m_{\Delta}$ | 0.01 | 0.001 | 0.0001 | 0.01 | 0.0001 |
| $M_{\Delta}$ | 0.1 | 0.01 | 0.001 | 0.1 | 0.001 |
| $m_{\mathbf{B}}$ | 0.0 | 0.0 | 0.0 | 0.0 | 0.0 |
| $M_{\mathbf{B}}$ | 0.5 | 1.0 | 1.25 | 0.25 | 1.25 |
| $m_{\mathbf{C}}$ | 0.0 | 0.0 | 0.0 | 0.0 | 0.0 |
| $M_{\mathbf{C}}$ | 0.75 | 1.0 | 0.5 | 1.0 | 0.5 |
| $m_{\mathbf{D}}$ | 0.5 | 0.0 | 0.75 | 1.0 | 0.0 |
| $M_{\mathbf{D}}$ | 1.0 | 0.25 | 1.0 | 1.0 | 1.0 |
| ridge $\alpha$ | 12.5 | 0.8 | 15.0 | 1.5 | 0.4 |
| MLP layers | 6 | 4 | 2 | 4 | 4 |
| batch $B'$ | 64 | 64 | 64 | 64 | 64 |
| lr | 0.001 | 0.0005 | 0.0005 | 0.0005 | 0.0005 |
| wd | 0.5 | 0.01 | 0.1 | 0.5 | 0.1 |
| rop | 0.2 | 0.2 | 0.2 | 0.2 | 0.2 |
| patience | 10 | 10 | 10 | 10 | 10 |

