# OpenReview forum: "Harnessing Untrained Dynamics: A Reservoir Computing Approach to State-Space Models"
_ICLR.cc/2026/Conference — ICLR 2026 Conference Withdrawn Submission_

### Official Review · Reviewer_58n7 · 2025-10-17

**Soundness:** 2
**Presentation:** 3
**Contribution:** 2
**Rating:** 4
**Confidence:** 5

**Summary:**

The paper proposes **Reservoir State Space Models (RSSMs)**: stacks of *fixed* (untrained) diagonal SSMs used as a reservoir, with only a light readout (ridge/MLP) trained. Using the convolutional view of SSMs, each block is executed via FFTs for O(L log L) parallel sequence processing. The theory restates stability and shows how (A, B, C, Δ) shape the convolution kernel. Nonlinearities between blocks create spectral “leakage” to increase expressivity. Empirically, RSSM is **very efficient** to train and competitive on sMNIST/pMNIST, but **significantly behind** trainable SSMs (e.g., S4) on harder tasks like sCIFAR-10 and several LRA tasks (which are still considered easy when compared to large-scale tasks).

**Strengths:**

* **Efficiency & simplicity.** Freezing the SSM core yields very high training throughput and low energy/CO₂.
* **Reasonable performance on easier/medium tasks.** Competitive or near-S4 on s/pMNIST; robust to permutation (pMNIST).
* **Reproducibility.** Few moving parts. Easy to reimplement and ablate.

**Weaknesses:**

* **Limited novelty.** The core idea is a frozen linear recurrent reservoir with a trained readout (Echo State Network/DeepESN), instantiated with SSM kernels and FFTs. This feels incremental relative to ESN and modern SSM work.
* **Underperforms on more challenging benchmarks.** Large accuracy gaps on sCIFAR-10 and several LRA tasks versus trainable SSMs (S4). This weakens the claim that the approach scales to difficult, long-range problems.
* **Claims vs. evidence mismatch.** The narrative suggests strong Path-X performance, yet the table shows S4 leading. **Please fix the Path-X discrepancy (text vs. table).**
* **Theory is mostly rederivation.** Stability bounds and kernel shaping follow directly from diagonal linear systems. No new approximation/capacity results that would justify accuracy trade-offs.
* **Ablations are insufficient where it fails.** We need deeper sensitivity on eigenvalue distributions, Delta, width/depth, and activation choices, especially on sCIFAR-10/LRA.
* **Comparative framing.** Missing compute-matched comparisons (equal time/FLOPs/params) against S4/other SSMs to more fairly quantify the accuracy–efficiency frontier.

**Questions:**

1. **Compute-matched comparison.** With equal wall-clock or FLOPs, how close can RSSM get to S4 on sCIFAR-10 and LRA?
2. **Sensitivity.** How sensitive are results to the sampling of lambda and Delta? Any principled sampler beyond uniform ranges (e.g., log-spaced decays/frequencies matched to task spectra)?
3. **Semi-trained variants.** If you learn per-channel scalings/mixers or a tiny convolutional mixer between blocks while keeping most of the reservoir frozen, how much of the accuracy gap closes at small cost?
4. **Nonlinearity choices.** Why ReLU for forward and tanh for readout features? Please ablate alternatives (e.g., GELU/SiLU; single shared nonlinearity).
5. **Streaming/online results.** Can you provide constant-time recurrent inference measurements on long streams to substantiate real-time claims?
6. **Fair baselines.** Were hyperparameters (e.g., regularization, data augmentations) tuned comparably for baselines, especially on sCIFAR-10?
7. **Clarification.** Please fix the Path-X discrepancy (text vs. table).

**Details Of Ethics Concerns:**

No concerns.

---

### Official Review · Reviewer_FMcH · 2025-10-21

**Soundness:** 2
**Presentation:** 2
**Contribution:** 2
**Rating:** 4
**Confidence:** 3

**Summary:**

This paper introduces the Reservoir State Space Model, a new neural architecture that merges state-space models with reservoir computing to efficiently model long-term dependencies in sequential data. Leveraging the linear structure of state-space dynamics, RSSMs implement convolutional operations that preserve internal states while enabling fast, parallelizable computation.

A key innovation is the use of untrained, structured convolutional dynamics as a fixed reservoir, with only a lightweight feedforward readout layer being trained. This design drastically reduces training cost and energy consumption while maintaining strong representational power. The authors conduct a stability analysis showing how system parameters govern memory capacity and dynamic behavior, and provide theoretical conditions for maintaining stable yet expressive reservoirs.

**Strengths:**

One clear advantage of this paper is its revival and modernization of the classical reservoir computing paradigm, which has long been recognized as a practical solution when recurrent weights are difficult or unstable to train. By integrating the structured dynamics of state-space models with untrained reservoirs, the authors demonstrate that powerful temporal representations can be achieved without backpropagation through time or complex recurrent optimization, a known bottleneck in recurrent neural networks

**Weaknesses:**

While the proposed approach is efficient and conceptually appealing, it remains unclear why the chosen initialization scheme leads to better results in general. The paper does not provide sufficient justification or analysis for how the initialization parameters influence performance beyond stability considerations. This raises questions about whether the method can consistently outperform standard training approaches when accuracy is prioritized. In particular, the large performance gap observed on tasks such as sequential CIFAR suggests that the proposed reservoir initialization may limit representational power or adaptability, especially in more complex settings. A deeper exploration or ablation on initialization choices would strengthen the empirical claims and clarify the trade-off between efficiency and predictive performance.

**Questions:**

It is unclear why the performance on sequential/permuted MNIST remains relatively stable across different configurations, while sequential CIFAR shows large variations. Could the authors clarify what factors contribute to this discrepancy? For example, is it due to differences in dataset complexity, feature dimensionality, or sensitivity of the reservoir initialization to higher-dimensional visual inputs?

---

### Official Review · Reviewer_U6nT · 2025-10-30

**Soundness:** 3
**Presentation:** 3
**Contribution:** 2
**Rating:** 4
**Confidence:** 3

**Summary:**

The authors propose an architecture that integrates state-space models (SSMs) with reservoir computing (RC) targeting long-term dependencies. They demonstrate the trade-off between accuracy, computational efficiency, and sustainability. Their approach aims to support real-world resource-constrained applications.

**Strengths:**

- The paper is of overall good quality.
- It follows a clear and logical structure.
- It shows promising results on several datasets using RC, demonstrating competitive accuracy (in Table 2) in addition to improvements in training time, CO2 emissions and energy consumption.
- It includes LRA experiments illustrating how the proposed approach compares to traditionally trained Transformers and S4.

**Weaknesses:**

- In Table 4, the only SSM presented is S4, which achieves the best performance, while the other models are transformer-based and the proposed RSSMs. Aside from showing that transformers perform poorly on these tasks, I don't see the benefit of highlighting the best and "second-best" scores, since the latter (if included) would likely come from other tuned SSMs. In my view, RSSMs don't become stronger simply because transformers perform worse.
- Minor issue: In the text, the authors refer to the S4 architecture as state-of-the-art, however, given the rapid pace of research on SSMs, this characterization is not really accurate anymore.
- Although the proposed approach has clear strengths in terms of training time, CO2 emissions and energy consumption, I am concerned about its practical impact and usability when the accuracy lags significantly behind trained models (with around a 40% drop in performance).

**Questions:**

- Which other state-of-the-art SSMs could be integrated into the RSSM design, and how generalizable is the proposed approach?
- Could the authors elaborate on how their method can still have a meaningful impact given the approximately 40% drop in accuracy in many  LRA tasks?

---

### Official Review · Reviewer_bBrt · 2025-10-31

**Soundness:** 2
**Presentation:** 3
**Contribution:** 2
**Rating:** 4
**Confidence:** 3

**Summary:**

The paper “Harnessing Untrained Dynamics: A Reservoir Computing Approach to State-Space Models” introduces the Reservoir State Space Model (RSSM), a hybrid neural architecture that merges State Space Models (SSMs) with Reservoir Computing (RC) to efficiently capture long-term dependencies in sequential data. RSSM leverages the structured linear dynamics of SSMs as a fixed, untrained convolutional reservoir, training only a lightweight feed-forward readout layer to drastically reduce computational cost and energy usage. The authors provide theoretical analyses of stability and representational capacity, demonstrating how the untrained dynamics yield rich, hierarchical temporal features. Empirical results on benchmarks such as sMNIST, psMNIST, sCIFAR-10, and Long Range Arena show that RSSM achieves accuracy comparable to state-of-the-art trainable models like S4 and GRU while offering superior efficiency and scalability, establishing it as a compelling approach for resource-constrained sequence modeling.

**Strengths:**

The paper’s strengths lie in its thoughtful synthesis, solid theoretical framing, and focus on efficiency and sustainability, an increasingly important dimension in modern machine learning. In terms of originality, the authors take a creative step by combining State Space Models (SSMs) and Reservoir Computing (RC), two areas that have evolved largely in parallel, into a unified architecture (RSSM) that inherits the stability and long-memory properties of SSMs while preserving the training simplicity of RC. Regarding quality, the paper provides a theoretical analysis of stability and representational capacity, connecting spectral properties of SSMs to memory and expressive power in deep untrained reservoirs. The experimental section, while not exhaustive, demonstrates a convincing efficiency–accuracy trade-off, showing that RSSM can approach the performance of trainable models such as S4 while requiring significantly less computation and energy. In terms of clarity, the paper is well organized, the derivations are accessible, and the diagrams clearly illustrate architectural design choices (e.g., removal of the mixing layer, layer stacking). Finally, in terms of significance, the work contributes to the growing discourse on sustainable and low-resource AI, offering an alternative path to high-efficiency sequence modeling without massive training overhead.

**Weaknesses:**

While the paper presents a clean synthesis of SSMs and reservoir computing, its novelty and relevance are somewhat limited given the current landscape. The proposed RSSM builds directly on ideas already explored in DeepESNs (Gallicchio et al., 2017) and diagonal SSMs like S4D (Gupta et al., 2022), but the paper does not convincingly explain how its fixed, untrained dynamics fundamentally improve upon or differ from these prior models. Moreover, the experimental comparisons are outdated: while S4 and GRUs are useful baselines, the field has moved toward more powerful and adaptive state-space architectures such as Mamba (Gu & Dao, 2024), which combine dynamic gating with learnable state updates. It remains unclear how an untrained reservoir model like RSSM competes with these modern approaches in both accuracy and scalability. Evaluating RSSM against Mamba or related architectures—or at least discussing whether the reservoir idea could be integrated into these adaptive frameworks—would make the contribution more relevant to ICLR 2026 audiences. In addition, the theoretical results, though well presented, mostly restate known stability and frequency-domain properties from SSM literature without empirically validating how they translate to improved modeling capacity or generalization. Finally, the efficiency and sustainability claims, while appealing, rely on coarse CO₂ estimates and do not provide standardized measurement conditions or normalized baselines. To strengthen the paper, the authors should include stronger contemporary baselines (e.g., Mamba, RWKV), analyze sensitivity to spectral initialization, and clarify how untrained structured reservoirs might complement or extend recent learnable state-space models.

**Questions:**

Please address the weaknesses above.

---

### Note · Authors · 2025-12-03

I have read and agree with the venue's withdrawal policy on behalf of myself and my co-authors.